# Aberrant non-canonical NF-κB signalling reprograms the epigenome landscape to drive oncogenic transcriptomes in multiple myeloma

Daniel A. Ang [1,10], Jean-Michel Carter [1,10], Kamalakshi Deka[1], Joel H. L. Tan [2], Jianbiao Zhou [3,4,5], Qingfeng Chen [2], Wee Joo Chng[3,4,5,6], Nathan Harmston [7,8,9] & Yinghui Li [1,2] ✉

In multiple myeloma, abnormal plasma cells establish oncogenic niches within the bone marrow by engaging the NF-κB pathway to nurture their survival while they accumulate pro-proliferative mutations. Under these conditions, many cases eventually develop genetic abnormalities endowing them with constitutive NF-κB activation. Here, we find that sustained NF-κB/p52 levels resulting from such mutations favours the recruitment of enhancers beyond the normal B-cell repertoire. Furthermore, through targeted disruption of p52, we characterise how such enhancers are complicit in the formation of superenhancers and the establishment of *cis*-regulatory interactions with myeloma dependencies during constitutive activation of p52. Finally, we functionally validate the pathological impact of these *cis*-regulatory modules on cell and tumour phenotypes using in vitro and in vivo models, confirming *RGS1* as a p52-dependent myeloma driver. We conclude that the divergent epigenomic reprogramming enforced by aberrant non-canonical NF-κB signalling potentiates transcriptional programs beneficial for multiple myeloma progression.

Multiple myeloma (MM) is a genetically complex and aggressive haematological malignancy arising from the uncontrolled proliferation of plasma cells in the bone marrow[1]. These antibody-secreting B lineage cells suppress normal hematopoiesis and produce abnormal M-proteins, leading to adverse clinical complications[2]. With an annual incidence of over 160,000 cases globally, MM is the second most prevalent blood cancer and accounts for >100,000 deaths per year[3,4]. Despite distinct improvements in the standard-of-care treatment and prognosis of MM patients over the past decade[5,6], it remains an incurable disease with an estimated median survival of 6 years[1]. One key challenge faced in the effective treatment of MM is the genetic heterogeneity of this disease[7,8]. Throughout the progression of MM

[1]School of Biological Sciences (SBS), Nanyang Technological University (NTU), 60 Nanyang Drive, Singapore 637551, Singapore. [2]Institute of Molecular and Cell Biology (IMCB), Agency for Science, Technology and Research (A*STAR), 61 Biopolis Drive, Proteos, Singapore 138673, Singapore. [3]Cancer Science Institute of Singapore, National University of Singapore, 14 Medical Drive, Centre for Translational Medicine, Singapore 117599, Republic of Singapore. [4]Department of Medicine, Yong Loo Lin School of Medicine, National University of Singapore, Singapore 117597, Republic of Singapore. [5]NUS Centre for Cancer Research, 14 Medical Drive, Centre for Translational Medicine, Singapore 117599, Singapore. [6]Department of Hematology-Oncology, National University Cancer Institute of Singapore (NCIS), The National University Health System (NUHS), 1E, Kent Ridge Road, Singapore 119228, Republic of Singapore. [7]Division of Science, Yale-NUS College, Singapore 138527, Singapore. [8]Program in Cancer and Stem Cell Biology, Duke-NUS Medical School, Singapore 169857, Singapore. [9]Molecular Biosciences Division, Cardiff School of Biosciences, Cardiff University, Cardiff CF10 3AX, UK. [10]These authors contributed equally: Daniel A. Ang, Jean-Michel Carter. ✉e-mail: liyh@ntu.edu.sg

from premalignant states, i.e. monoclonal gammopathies of undetermined significance (MGUS) and smouldering multiple myeloma (SMM), neoplastic B cells accumulate various primary and secondary genomic abnormalities[9,10]. These genetic alterations contribute to the deregulation of gene expression programmes that correlate with disease severity and relapse[11–13].

Major efforts to stratify the prognosis and treatment of MM patients based on genomic profiles and expression subtypes have yielded promising insights[14–18]. However, more work dissecting the gene regulatory mechanisms altered by driver mutations and understanding their impact on the myeloma transcriptome is necessary to accelerate the development of effective, personalized medicine. Recent epigenome studies have highlighted the distinct changes that occur in the chromatin landscape during myelomagenesis and their critical involvement in the activation of oncogenic transcriptional programmes[19–22]. In particular, enhanced chromatin accessibility and decompaction of heterochromatin at distal regulatory elements have been correlated with the aberrant activation of de novo enhancers and super-enhancers (SEs) in myeloma plasma cells[19–23]. Such MM specific enhancers and chromatin alterations regulate oncogenic gene expression signatures that have been linked to distinct genetic subtypes and clinical outcomes[19–24].

Chromatin plasticity and enhancer dynamics have been increasingly implicated in cancer progression[25–28]. However, the precise mechanisms of chromatin regulation and transcriptional activities at altered enhancers in MM remains elusive. Nuclear factor κB (NF-κB) is one of the transcription factors that plays a critical role in the survival and proliferation of MM[29,30]. Both the classical and non-canonical NF-κB pathways are frequently activated in malignant plasma cells due to diverse genomic lesions and signals from the tumour microenvironment[31,32]. Similar to normal plasma cells, more than 80% of MM tumours hyperactivate NF-κB in response to ligands, such as BAFF and APRIL, present in the bone marrow microenvironment[33–35]. However, 15–20% of MM cases acquire genetic alterations in key regulators of the signalling cascade, leading to sustained NF-κB activity that is independent of the microenvironment[36–38].

Notably, recurrent mutations that result in persistent activation of the non-canonical NF-κB pathway are preferentially enriched in MM relative to other B-cell malignancies[30,39,40]. These genomic abnormalities occur during the transition from MGUS to MM, suggesting that aberrant activation of the non-canonical NF-κB pathway and increased autonomy from the microenvironment are key steps in the malignant progression of MM[37,38,41].

In this study, we investigate the role of constitutive non-canonical NF-κB activation in the myeloma epigenome and characterize the genomic landscape of NF-κB/p52 binding in MM cell lines that harbour genetic dysregulation of non-canonical NF-κB signalling. We identify diverse enhancer states that are associated with NF-κB/p52 in myeloma plasma cells and analyse their epigenomic changes upon loss of constitutive p52 activation. The p52-dependent changes in chromatin accessibility are shown to correlate with the activation of super-enhancers and an oncogenic transcriptome in MM. We further reveal the regulation of myeloma essential genes via chromatin interactions driven by p52 dependent super-enhancers. Collectively, our data suggests a model whereby mutational activation of the non-canonical NF-κB pathway contributes to rewiring of the myeloma epigenome through p52-driven enhancer co-option which sustains oncogenic transcriptional programmes.

## Results

### Constitutive NF-κB activation is associated with oncogenic transcriptomes in MM subtypes

To identify the oncogenic transcriptional programmes associated with genomic alterations and constitutive activation of the NF-κB pathway in MM, we first analysed the mutational and transcriptional profiles of

629 newly diagnosed patient samples from the MMRF CoMMpass study (NCT01454297). We classified the level of NF-κB transcriptional activity among tumour samples using an NF-κB expression index that was defined based on the geometric mean expression of 11 NF-κB target genes[37]. In parallel, we obtained mutation profiles for 11 genes encoding regulators of the NF-κB pathway that are susceptible to gain or loss of function mutations (see Methods)[36–38]. We divided the MM patients into two groups displaying higher or lower NF-κB indices: NF-κB+ or NF-κB- respectively (see Methods). Past studies have documented the prevalence of mutations contributing to hyperactivation of the non-canonical NF-κB (ncNF-κB) pathway[37,38]. Consistent with this, we detected a high frequency (32%) of such mutations with NF-κB + patients being 3.5 times more likely to carry a coding mutation in one of the 11 regulatory genes (Fisher's Exact Test, $p = 9.583 \times 10^{-10}$; Fig. 1a). In particular, 82% of the patients carrying *TRAF3* mutations were classified in the NF-κB+ group, re-affirming the inactivation of this tumour suppressor as a major driver of constitutive ncNF-κB activity in MM[38]. The remaining NF-κB+ tumours lacking mutations are likely relying on either signals from the bone marrow microenvironment or undocumented mutations/mechanisms to engage the NF-κB pathway[42].

We further sought to identify the subtypes that were enriched in NF-κB+ patients based on the RNA expression subtyping recently performed using the CoMMpass dataset[18]. Interestingly, three out of the four subtypes showing worse overall survival in MM patients (*MAF*: t(14;16), *HRD low TP53*: hyperdiploid with low *TP53* expression and *1q gain*: gain of chromosome 1q—further details in[18]) were found to be significantly enriched in NF-κB+ samples (Fig. 1b, Supplementary Fig. 1a). *1q gain* in particular has been shown to be associated with worse prognosis in newly diagnosed and relapsed MM patients[43–46]. This suggests that constitutive NF-κB activity may be impacting survival in these subtypes and is consistent with earlier reports linking NF-κB signalling to increased cancer severity[30,47]. Indeed, deletions affecting *TRAF3* have been reported to be enriched in the *1q gain* subtype[18]. In contrast, two subtypes were significantly depleted in NF-κB+ samples, a low NF-κB HRD karyotype subtype and the proliferative (PR) subtype, which may represent tumours that display alternative mechanisms or have escaped their dependence on the NF-κB pathway.

We next performed a differential expression analysis between the NF-κB+ and NF-κB- transcriptomes from the CoMMpass study and identified the biological processes that are associated with these distinct groups of patients. Despite the general heterogeneity of MM tumours, 6576 genes were found to be significantly upregulated (FDR < 0.1) in the NF-κB+ group[8] (Supplementary Fig. 1b). As expected, NF-κB+ upregulated genes and associated subtypes were prominently enriched in cancer-promoting programmes including angiogenesis, cell migration/motility, cell adhesion, mitogen-activated protein kinase (MAPK) and Wnt signalling cascades, along with the NF-κB pathway (Supplementary Fig. 1c and d). In contrast, NF-κB+ downregulated genes were associated with cellular functions including DNA replication/elongation, DNA repair/homologous recombination and metabolic processes (Supplementary Fig. 1e).

To further characterise the p52-dependent transcriptional programmes active in NF-κB+ multiple myeloma, we disrupted the constitutive expression of p52 in several NF-κB+ Multiple Myeloma cell lines (MMCL) by targeting the Rel Homology Domain (RHD; Fig. 1c) of the *NFKB2* gene (*NFKB2* KD). The procedure most effectively reduced p100 expression and further processing to p52 in KMS-11, without affecting components of the classical NF-κB pathway including p105/p50, RELA and c-REL (Fig. 1d and Supplementary Fig. 1f). Following CRISPR/Cas9-targeted p52 disruption in NFKB2 KD cells, RNA-seq analysis revealed the significant downregulation of 1263 genes (FDR < 0.1; 1099 in MM1.144) (Fig. 1e and Supp. Fig. 1g). Many of the downregulated genes included those identified by the Dependency Map (DepMap) project as essential for the survival and proliferation of

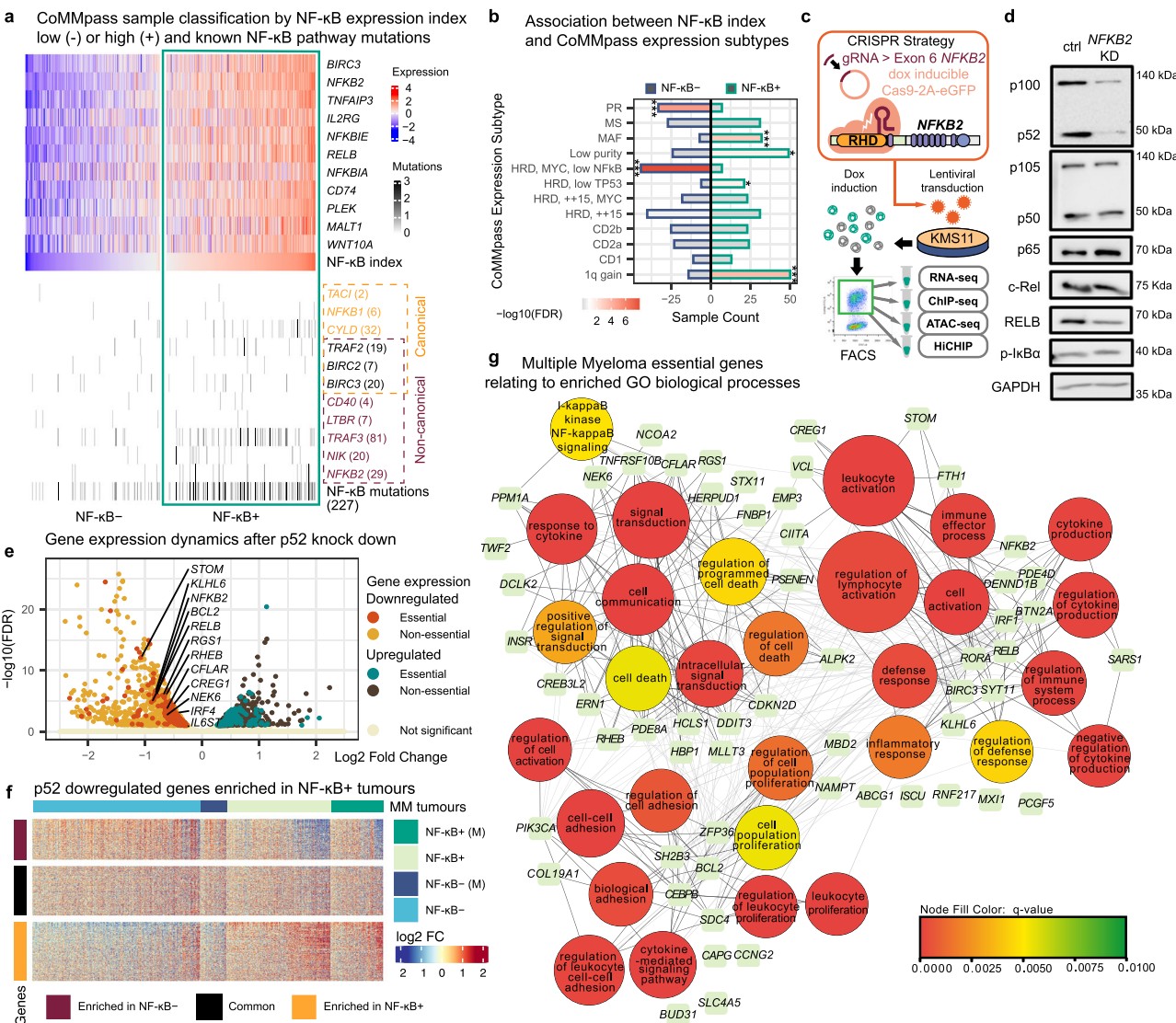

**Fig. 1 | Constitutive NF-κB pathway activation is associated with oncogenic transcriptional programmes in aggressive MM subtypes. a** Classification of CoMMpass patient samples using an NF-κB index based on the geometric mean of the expression for 11 NF-κB signature genes. Samples with a positive NF-κB index and expression beyond the upper quartile for at least 1 contributory gene are classified as NF-κB + (red) and the remaining as NF-κB- (blue). Mutations in 11 genes involved in NF-κB pathway activation are also shown. **b** Counts of samples belonging to one of the 12 RNA subtypes identified in[18] and one of the two NF-κB groups. Bar colours indicate the confidence level (white to red: < 0.1 FDR, white to grey: > 0.1 FDR) for a negative or positive association for each subtype with NF-κB+ samples (*FDR < 0.05; ***FDR < 0.001; *n* = 583, Fisher's Exact Test with Benjamini & Hochberg adjustment). **c** CRISPR design and experiment strategy to target *NFKB2* expression. The guide RNA (gRNA) targets the Rel homology domain (RHD) of *NFKB2* introducing a nonsense mutation. KMS-11 cells are transduced with CRISPR targeting plasmids, induced with doxycycline (Dox) for Cas9 expression and sorted by flow cytometry to harvest Cas9/gRNA-expressing clones for downstream next-generation-sequencing (NGS) experiments. **d** A representative western blot showing changes to factors involved in the canonical (NFKB1: p105/p50, p65, c-Rel, p-IκBα) and non-canonical NF-κB (NFKB2: p100/p52, RelB) pathways upon CRISPR-Cas9 knockdown of *NFKB2* in KMS-11, *n* = 2. **e** Differentially expressed genes (DEGs) identified following *NFKB2* knockdown (downregulated in yellow; upregulated in brown), with essential genes highlighted (downregulated in red; upregulated in green) and named. Non-significant regulation in pale yellow. **f** Heatmap integrating downregulated genes from KMS-11 CRISPR cells with DEGs identified in NF-κB+ samples from CoMMpass study. Samples exhibiting mutations identified in the NF-κB pathway are indicated (M). **g** Network visualisation of biological process terms associated with essential genes downregulated after p52 knockdown (*Q*-value < = 0.01) and enriched in tumours forming three major clusters.

myeloma cells[48] (Supp. Fig. 1h). Among the essential genes, we observed downregulation of known NF-κB target genes such as *BCL2* and *IRF4* (particularly in KMS-11), which have been reported to mediate apoptotic resistance in MM cells[49-52] (Fig. 1e and Supp. Fig. 1g).

Integration of NF-κB+ patient expression data showed that 35% of genes downregulated in the *NFKB2* KD KMS-11 transcriptional profile were overexpressed in NF-κB+ primary MM tumours (Fig. 1f). Consolidating KMS-11 as our primary experimental model for investigating the role of p52 in multiple myeloma, we identified several p52-regulated essential genes upregulated in patients, i.e. *RGS1* and *RHEB*, whose expression have been linked to poorer prognosis in MM

patients and other cancers[53,54]. To visualise the critical transcriptional programmes regulated by these genes, we performed a network projection of the major biological processes that they are associated with. Clustering of the network revealed three major themes relevant for myelomagenesis (1) regulation of apoptosis, (2) inflammation or lymphocyte activation, (3) cell proliferation and adhesion (Fig. 1g). p52-dependent transcriptional programmes involved in the inflammatory response (2) appeared closely intertwined with pro-oncogenic programmes (1 and 3)[55]. Recent studies have identified a role for NF-κB dynamics in activating latent enhancers to regulate gene expression changes during stimuli induction[56,57]. We hypothesized that these

oncogenic programmes may be activated at a *cis*-regulatory level during aberrant NF-κB/p52 signalling as the p52-associated transcriptional complexes are deployed beyond normal thresholds for plasma cell maintenance[58,59].

## NF-κB/p52 binding is associated with MM dormant enhancer activation

To gain insights into the *cis*-regulatory mechanisms that may be driving such oncogenic transcriptional programmes during aberrant ncNF-κB signalling, we investigated the NF-κB2/p52 occupancy at active enhancers of MM. We profiled p52 binding and H3K27 acetylation in a panel of MM cell lines bearing genetic hyperactivation of the pathway (NF-κB + MMCLs) using chromatin immunoprecipitation sequencing (ChIP-seq). MM1.S, MM1.144, U266, LP1, KMS-11 and JJN3 contained clinically relevant mutations in *TRAF3* and *NIK*[38]. Consistent with earlier observations of aberrant ncNF-κB activity arising from these mutations[37,38], NF-κB + MMCLs displayed elevated

NIK protein levels and constitutive processing of NF-κB2 p100 – p52, in comparison to non-mutant MMCLs (XG7, H929 and MOLP8) (Fig. 2a). We obtained 27489 unique p52 bound sites across the genome and identified the expected NF-κB2 motif centred across peaks (Fig. 2b and Supp. Fig. 2a–c). The binding sites were primarily detected in intergenic (29%) and intronic (42%) regions, which are indicative of enhancer colocalization and corroborate previous findings[60] (Fig. 2b, c). Meanwhile, a smaller proportion (21%) of binding sites were located at promoter regions (Fig. 2b). Additionally, we observed a significant enrichment of strong enhancer states at p52 binding sites in MM relative to normal plasma cells and other B-cell developmental stages (FDR < 0.05; 26% of p52 binding sites overlap with a strong enhancer ChromHMM state)[21,61] (Supp. Fig. 2d). This suggested that a significant amount of p52 binding sites were associated with *cis*-regulatory modules involved in the transition from other chromatin states to active enhancer states during myelomagenesis (Fig. 2d).

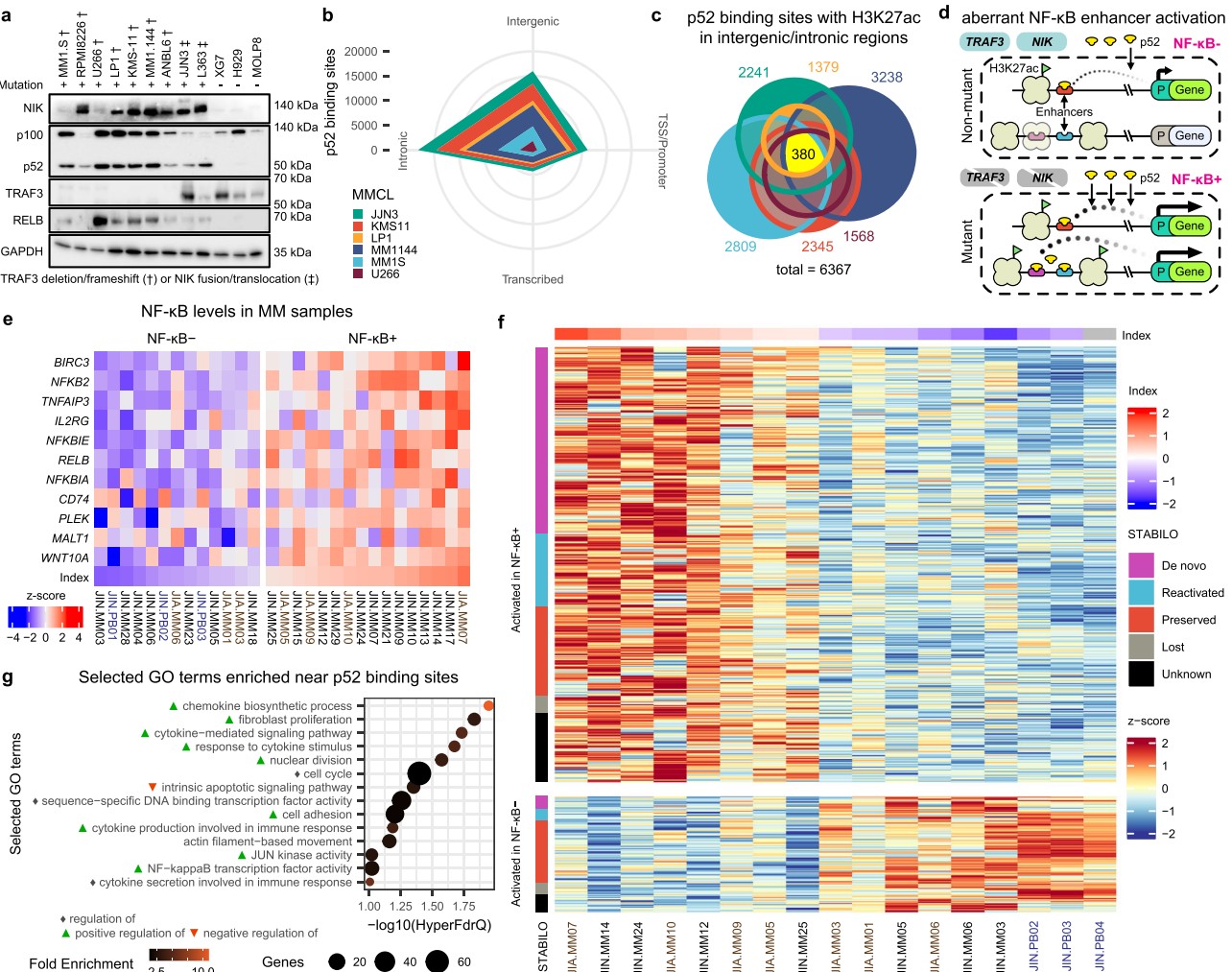

**Fig. 2 | Aberrant NF-κB signalling targets dormant myeloma enhancers.**
**a** Western blot for key non-canonical NF-κB pathway members - NFKB2 (p100/p52), NIK, TRAF3 and RelB across constitutively activated mutant MMCLs (MM1.S, RPMI8226, U266, LP1, KMS-11, MM1.144, ANBL6, JJN3, L363) resulting from *TRAF3* and *NIK* mutations as well as non-mutant MMCLs (XG7, H929, MOLP8), *n* = 2. **b** Radar plot showing distribution of detected p52 binding sites at genomic features. **c** Euler diagram illustrating the overlap in p52 binding sites (*n* = 6367) at intergenic or intronic H3K27ac peaks (peaks in at least two replicates) detected across all six MMCLs with total sites in each cell line indicated. **d** Proposed mechanism for the activation of dormant enhancers during hyperactive NF-κB

signalling in NF-κB mutant myeloma cells. **e** Classification of patient MM and PB samples (blue text) using an NF-κB index based on the geometric mean of the expression for 11 NF-κB signature genes (as performed for the MMRF data). RNA-seq datasets are matched with epigenomic data from two main studies[20] and[19] annotated in brown and black text respectively. **f** H3K27ac signal (Z-score of normalised rLog counts) at intronic and intergenic p52 binding sites across multiple myeloma patients contrasting high and low NF-κB index samples. Loci are annotated with STABILO. **g** Cancer related biological processes found to be significantly associated with genes in proximity to the p52 binding sites enriched in H3K27 acetylation in NF-κB+ patients relative to NF-κB- (GREAT analysis; HyperFdrQ ≤ 0.1).

We defined 6367 putative enhancers (intergenic/intronic H3K27ac peaks) showing p52 binding potential (≥2 replicates) across the MMCL panel we tested (Fig. 2c). Putative enhancers were confirmed to colocalise with H3K4me1 peaks in KMS-11 and were devoid of H3K27me3 marks (Supp. Fig. 2e). Enhancer landscapes were generally heterogenous between MMCLs with only 380 enhancers consistently identified across MMCLs. To further characterize the chromatin states of p52-bound enhancers in MM, we classified enhancer activity based on their chromatin state (ChromHMM) transitions across B-cell development[61,62]. We term this as the STABILO (State Transitions Across B-cells Imputed Locus Origin) classification comprising of five distinct enhancer origins: MM specific enhancers that are newly formed in tumours (de novo); MM enhancers previously activated in B-cell development but absent in plasma cells (reactivated); MM enhancers previously activated in plasma cells and optionally in B-cell development (preserved); Enhancers previously activated in B-cell development or plasma cells but not in MM (lost); Enhancers not previously associated with active enhancer marks (unknown) (see Methods and Supp. Fig. 2f). The majority of the putative enhancers identified (87%) were activated in at least one of the NF-κB+ MMCLs when compared to NF-κB- MMCLs, H929 or KMS-28BM (Supp. Fig. 2g). Among these NF-κB+ MMCL enriched enhancers, 1711 were classified as de novo or reactivated (Supp. Fig. 2g). These observations indicated that a significant proportion of p52-bound enhancers activated during myelomagenesis (31%) could have been potentially co-opted from enhancers usually active in B cells or other lineages, which we refer to collectively as dormant enhancers.

To determine whether the p52 enhancer landscape is similarly activated in primary tumours from MM patients, we used 26 MM and 3 plasmablast (PB) samples for which H3K27ac ChIP-seq or ATAC-seq data were available and performed complementary RNA-seq where necessary to group these samples by NF-κB index (as performed for the MMRF data) (Fig. 2e)[19,20]. We identified 470 p52 loci located in intergenic or intronic regions with consistently elevated H3K27 acetylation in NF-κB+ MM samples (Fig. 2f). Notably, the H3K27ac-enriched p52 loci were frequently associated with de novo or reactivated enhancers in NF-κB+ MM tumours (50.34%) compared to MMCLs (Fig. 2f and Supp. Fig. 2g). Gene ontology analysis for genes within the proximity of intergenic/intronic p52 loci showed a significant enrichment in biological processes related to the oncogenic transcriptional programmes highlighted previously (Fig. 2g).

## NF-κB/p52 dependent chromatin remodelling involves dormant enhancers in MM

Enhancer activation in MM has recently been shown to regulate genes critical for MM progression[19,20,22]. To elucidate the link between the p52 cis-regulatory landscape and the activity of genes participating in oncogenic transcriptional programmes in MM, we profiled the H3K27ac and chromatin accessibility changes upon NFKB2 KD in KMS-11 cells and integrated it with patient epigenomic data (Fig. 3a–c and Supp. Fig. 3a–e). Principal component analysis suggested distinct and consistent changes in the enhancer landscape following our p52 knockdown strategy (Supp. Fig. 3c). Depleted H3K27ac peaks showed increased p52 co-occupancy compared to gained H3K27ac peaks, followed by concordant changes in accessibility (Fig. 3a, b). Specifically, 889 (31%) of the 2868 downregulated H3K27ac peaks, compared to 180 (7%) of the 2606 upregulated peaks, overlapped p52 binding sites detected in KMS-11 cells (Supp. Fig. 3a). Loss of H3K27ac signal was correlated with the loss of accessible chromatin at p52 bound regions (Fig. 3b and Supp. Fig. 3e). However, chromatin accessibility only showed minor changes at p52 bound loci with increased H3K27ac signal (Fig. 3b). Differential H3K27ac peaks were mainly located at intronic and intergenic regions (Supp. Fig. 3d). These data indicated that p52 dependent chromatin remodelling is primarily associated with enhancer activation in MM. This is consistent with previous

reports linking NF-κB activity to increased chromatin accessibility and enhancer function[56,57].

Expression of the genes nearest to non-promoter H3K27ac peaks closely mirrored the changes in enhancer activity, indicating the p52 knockdown was effectively impacting enhancer driven cis-regulatory mechanisms of transcription (Supp. Fig. 3f). All the enhancers lost upon NFKB2 KD were previously identified as active in the mutant MMCL panel. Lost enhancers were also more likely to be of de novo or reactivated class (dormant enhancers; $p < 2.2 \times 10^{-16}$ Supp. Fig. 3g). The p52-dependent H3K27ac peaks identified in KMS-11 showed similar H3K27 acetylation and accessibility dynamics in MM1.144 during p52 KD (Supp. Fig. 3h). Additionally, 477 showed consistent enrichment in H3K27 acetylation or accessibility in NF-κB+ patients (Fig. 3c). The majority of these loci appeared to be dormant enhancers (Fig. 3c). Once again, expression of genes nearest to p52-dependent NF-κB+ enriched enhancers showed similar trends using both matched patient NF-κB + RNA-seq and MMRF datasets (Fig. 3d, e). Several of the NF-κB+ samples showed evidence of ncNF-κB activating mutations affecting CYLD and TRAF3 (Fig. 3f, g), highlighting that such enhancer and promoter dynamics are likely associated with genetically induced aberrant p52 levels. Collectively, these results suggest that constitutive activation of ncNF-κB impacts the cis-regulatory programming of multiple enhancers unique to MM to contribute to oncogenic transcription, besides maintaining existing plasma cell enhancers in myeloma cells.

## NF-κB/p52 enhancer reprogramming impacts super-enhancer dynamics linked to aberrant expression of myeloma essential genes

We then evaluated the extent to which p52-dependent enhancers mediate oncogenic transcription in MM. Given the significant role of super-enhancers (SEs) in regulating myeloma transcriptomes and the ability of NF-κB to coordinate SE formation[19,20,56], we analysed the SE profiles based on H3K27 acetylation in NFKB2 KD KMS-11 cells and primary MM tumours. Numerous SEs were concurrently downregulated alongside proximal MM essential genes (DepMap) following p52 KD (Fig. 4a)[48]. At least 30% of p52-dependent constituent enhancers (CEs) within the downregulated SEs overlapped p52 binding sites (Fig. 4b). Interestingly, a large proportion of the p52 bound CEs which contribute to such SE dynamics were classified as reactivated or de novo and featured enriched H3K27 acetylation or accessibility in NF-κB+ MM tumours (relative to NF-κB- PB) (Fig. 4b).

Closer inspection of MM essential genes of clinical relevance, such as BCL2 and IL6ST[50,53,63], revealed p52 bound SEs in their vicinity. These SEs displayed loss of chromatin accessibility or H3K27 acetylation in CEs following p52 KD (Fig. 4c, d and Supp. Fig. 4). These p52-dependent enhancers also displayed stronger H3K27 acetylation or accessibility in NF-κB+ MM samples on average, correlating with the increased average expression of the neighbouring genes measured in the MMRF NF-κB+ samples, particularly in the 1q gain subtype (Fig. 4e, f). Such changes suggest that the SE dynamics regulated by p52 play a critical role in the aberrant expression of myeloma essential genes as well as generally impacting nearby gene expression (Supp. Fig. 5). Furthermore, ATAC-seq footprinting analyses effectively confirmed the increased binding activity of p52 and several other related transcription factors including AP-1 and IRF family members (Fig. 4g, h and Supplementary Fig. 4b-e), indicating that constitutive ncNF-κB activity may directly or indirectly be modulating SE potency. Altogether, these data demonstrate that mutational activation of the ncNF-κB pathway is associated with dormant enhancer activation alongside preserved enhancers to potentiate SEs in MM.

## p52 mediated super-enhancer reprogramming drives chromatin interaction changes targeting the myeloma transcriptome

Regulatory elements impacting MM have been demonstrated to activate the expression of target genes through enhancer-promoter cis-

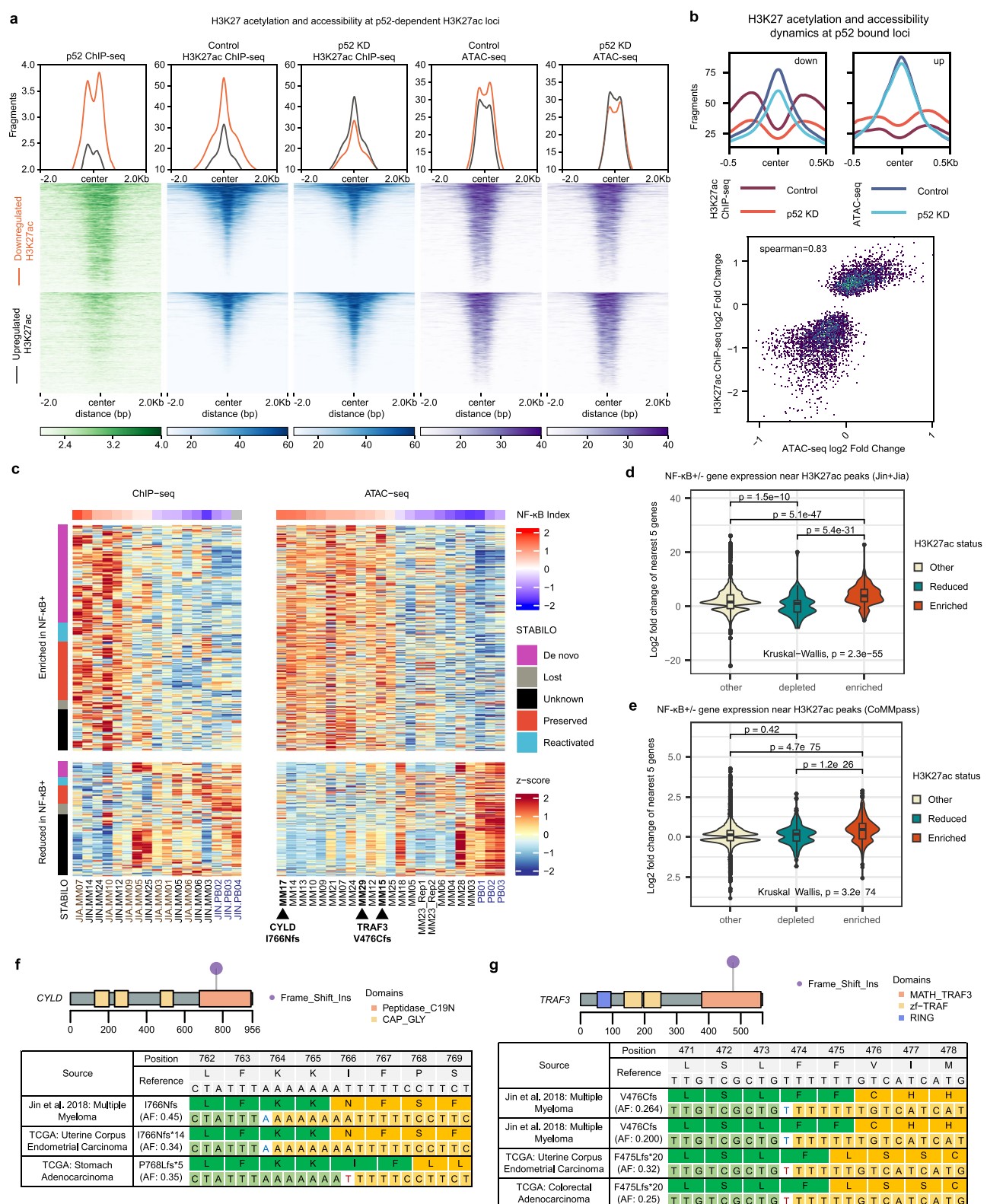

regulatory interactions[21]. To determine whether the p52-dependent typical enhancers (TEs) and SEs identified were interacting with potential oncogenes, we performed H3K27ac HiChIP to capture the enhancer interactome in KMS-11 cells with and without disruption of p52. From the 44476 significant chromatin loops (>10 KB to 2 MB) called across the two conditions, we detected 30651 (69%) to be differentially regulated, of which two thirds showed decreased contact counts following p52 disruption (Fig. 5a and Supp. Fig. 6a, b). We also

noted limited changes in topologically associating domains (TADs) with 21% showing significant inter-TAD or intra-TAD changes (Supp. Fig. 6c).

The differential loops were then assigned for putative Enhancer-Enhancer (EEI), Enhancer-Promoter (EPI) or Promoter-Promoter (PPI) interactions (see Methods for details). We found that the majority of chromatin interaction changes were EPIs ($n = 39443$; 46%) followed by EEIs (33%) and PPIs (21%) (Supp. Fig. 6d). EPIs were also the most

**Fig. 3 | H3K27 acetylation and accessibility dynamics after NFKB2 knock down reveal putative p52-dependent enhancers. a** Loci displaying differential H3K27 acetylation and accessibility as well as p52 binding signal in KMS-11 cells. **b** Downregulated or upregulated H3K27 acetylation and accessibility signals, centred on p52 binding sites throughout the genome, in KMS-11 control (ctrl) versus p52 KD cells. Scatter plot showing correlation between H3K27 acetylation and accessibility dynamics at p52 binding sites following p52 disruption. **c** H3K27 acetylation and MM accessibility signal dynamics across p52-dependent H3K27ac peaks identified in KMS-11. **d** Matched RNA-seq and **e** MMRF expression dynamics of nearest 5 genes to p52-dependent H3K27ac peaks found to have enriched accessibility or H3K27 acetylation in NF-κB+ MM samples. 3 groups of peak/gene pairs are defined: Depleted ($n$ = 1747), Enriched ($n$ = 2452) and Other ($n$ = 213192). Lower and upper hinges correspond to first and third quartiles. Central value corresponds to the median. Whiskers extend to largest/smallest values no further than 1.5 x IQR (Interquartile range). Pairwise-comparison $p$-values determined by 2-sided Wilcoxon rank sum tests and adjusted for multiple comparisons (Benjamini-Hochberg). **f** Somatic *CYLD* frameshift insertion putatively detected in multiple myeloma sample MM17 from RNA-seq data. **g** Somatic *TRAF3* frameshift insertion putatively detected in multiple myeloma samples MM15 and MM29 from RNA-seq data. Similar somatic mutations recorded in other cancers are shown based on TCGA data. Such mutations are implicated to contribute to constitutive activation of the ncNF-κB pathway.

attenuated *cis*-regulatory interactions following p52 knockdown (Fig. 5b and Supp. Fig. 6d). This indicated that constitutive ncNF-κB signalling has a significant role in promoting active *cis*-regulatory interactions between enhancers and gene promoters.

To comprehensively map the changes in EPIs with oncogenic transcriptional programmes that are relevant in MM patients, we integrated the enhancer and gene expression dynamics identified in KMS-11 and clinical samples with the p52 dependent interactome. This analysis revealed a positive correlation between enhancer dynamics and promoter-associated gene expression pairs as defined by each interaction (Fig. 5c). 1156 enhancer-promoter interaction (EPI) pairs were found to display significant epigenomic changes associated with *cis*-regulatory interactions, enhancer, and promoter dynamics (Supp. Fig. 6c). The majority of the attenuated chromatin interactions (76%) were associated with decreased enhancer and promoter activity after p52 knockdown (Supp. Fig. 6e and Supp. Table 2). Despite the notable change in TAD stability following p52 knock down, most of these EPIs were associated with stable TADs (Supp. Fig. 6f). Furthermore, 69% of these EPIs showed enriched epigenomic activity and/or gene expression in NF-κB + MM tumours, especially for those bound by p52 at their enhancers (80%) (Fig. 5c). These observations demonstrate that aberrant activation of the non-canonical NF-κB pathway can potentially activate the myeloma transcriptome through EP *cis*-regulatory modulations.

Among the EPIs linking features with significant dynamics, 330 lead to significant downregulation in target gene activity in KMS-11 upon p52 KD. The majority of these involved SEs or TEs that were lost following p52 knockdown (309; 94%), of which 238 (77%) showed increased activity in NF-κB+ samples. Finally, 184 (77%) of these EPIs were also associated with MMRF NF-κB+ enriched genes (Fig. 5d). Interestingly, p52 downregulated genes linked to EPIs that displayed enriched epigenomic and/or transcriptional activity were frequently associated with dormant (de novo/reactivated) enhancers (47%) of which 54% could be contributing to super-enhancers (Supp. Fig. 6g). This suggested that deregulated ncNF-κB signalling can induce enhancer/SE formation or co-opt developmental enhancers that are pre-established to activate key genes in myelomagenesis through *cis*-regulatory interactions. Our current data indicates p52-dependent enhancers can exert their influence through proximal or distal interactions or indeed both with seemingly additive effects on gene transcription (Supp. Fig. 7). In summary, through these integrative analyses, 16 myeloma essential genes (including *BCL2*) were found to be targeted by p52 bound enhancers showing increased activity in NF-κB+ samples. Dormant enhancers were implicated in nine of these essential genes such as *BCL2*, *IL6ST* and *RGS1* (Fig. 5e)[48].

To cast light on the possible non-myeloma functions or origins of the p52-bound regulatory elements classified as de novo enhancers, we examined the activity of these loci across the Roadmap Epigenomics collection which encompasses a diverse array of tissue types. We found that these sites were heterogeneously activated across different non-lymphoid tissues. The lymphoid tissues generally also followed this trend, indicating that this enhancer subset did not feature enhancers that were simply reverting to lymphoid-related roles.

Strikingly, the GM12878 lymphoblastoid B-cell line which is known to constitutively activate the NF-κB pathways[64] differed from the rest of the samples and showed increased activity in the majority of the loci selected (55%) (Fig. 5f). This suggests constitutive p52 activity is able to unlock a diverse repertoire of enhancers that are usually selectively activated in other tissues.

## p52 super-enhancer reprogramming impacts the expression and oncogenic activity of MM dependencies

To evaluate whether dynamic EP pairs targeted by p52 transcriptional complexes are critical for activating oncogenic transcriptomes, we prioritised genomic loci that contain p52 bound, dynamic SEs and myeloma essential genes regulated by p52. We selected SEs putatively targeting *BCL2* and *RGS1*; two candidate myeloma essential genes that displayed a strong enrichment in chromatin accessibility and/or H3K27ac signal in NF-κB+ tumours (highlighted in yellow in Fig. 6a and Supp. Fig. 8) as well as increased expression in NF-κB+ MMRF patients (Fig. 6b). Interestingly, the *RGS1* and *BCL2* SEs featured p52 dependent constituent enhancers classified as de novo or reactivated enhancers, besides other enhancers (Fig. 6a and Supp. Fig. 8). We hypothesised these SEs could be potentiated by the reactivation of such dormant enhancers in response to p52 binding during constitutive NF-κB signalling and subsequently impacting the putative target gene expression in myeloma cells.

To investigate this further, we first verified these p52-bound constituent enhancers were effectively capable of enhancing transcription using a luciferase assay (Supp. Fig. 9a). We then introduced deletions to them in MMCLs using CRISPR-Cas9 and examined the effect of these deletions on target gene expression. All MMCLs were transduced with CRISPR guide RNAs (gRNAs) targeting p52 binding sites at the enhancers, together with Cas9, and the genomic deletions were verified by Sanger sequencing (see Supp. Fig. 9b for example). The preserved constituent enhancer in the *BCL2* SE was first deleted in KMS-11, leading to an attenuation in the expression of BCL2 (Supp. Fig. 9c, d). This particular oncogene is a major therapeutic target in haematological malignancies[65,66] which has been linked to NF-κB in prostate cancer and EBV-induced immortalisation of B cells[49,67]. However, the nature of the *cis*-regulatory relationship between p52 and *BCL2* had been missing in MM.

Deletion of the de novo constituent enhancer in the *RGS1* SE (Supp. Fig. 9e) also attenuated RGS1 expression in KMS-11 and this effect was confirmed in other NF-κB + MMCLs (LP1 and JJN3) (Fig. 6c). Several groups have also reported the overexpression of *RGS1* transcript in various cancer tissues including breast cancer, cholangiocarcinoma, oesophageal carcinoma, glioblastoma multiforme, lung adenocarcinoma, and stomach adenocarcinoma[68,69]. Although the R4 family of RGS proteins have been reported to display inhibitory roles in cell migration and trafficking in non-cancer cells[70–72], various RGS proteins have been positively correlated with cancer migration and invasion in certain cancer types[73–77]. Strikingly, we found that proliferation, adhesion and survival of MM cells were significantly downregulated upon *RGS1* SE deletion, mirroring the results from shRNA knock down of *RGS1* in MM cells (Fig. 6d–f and Supp. Fig. 9f–i).

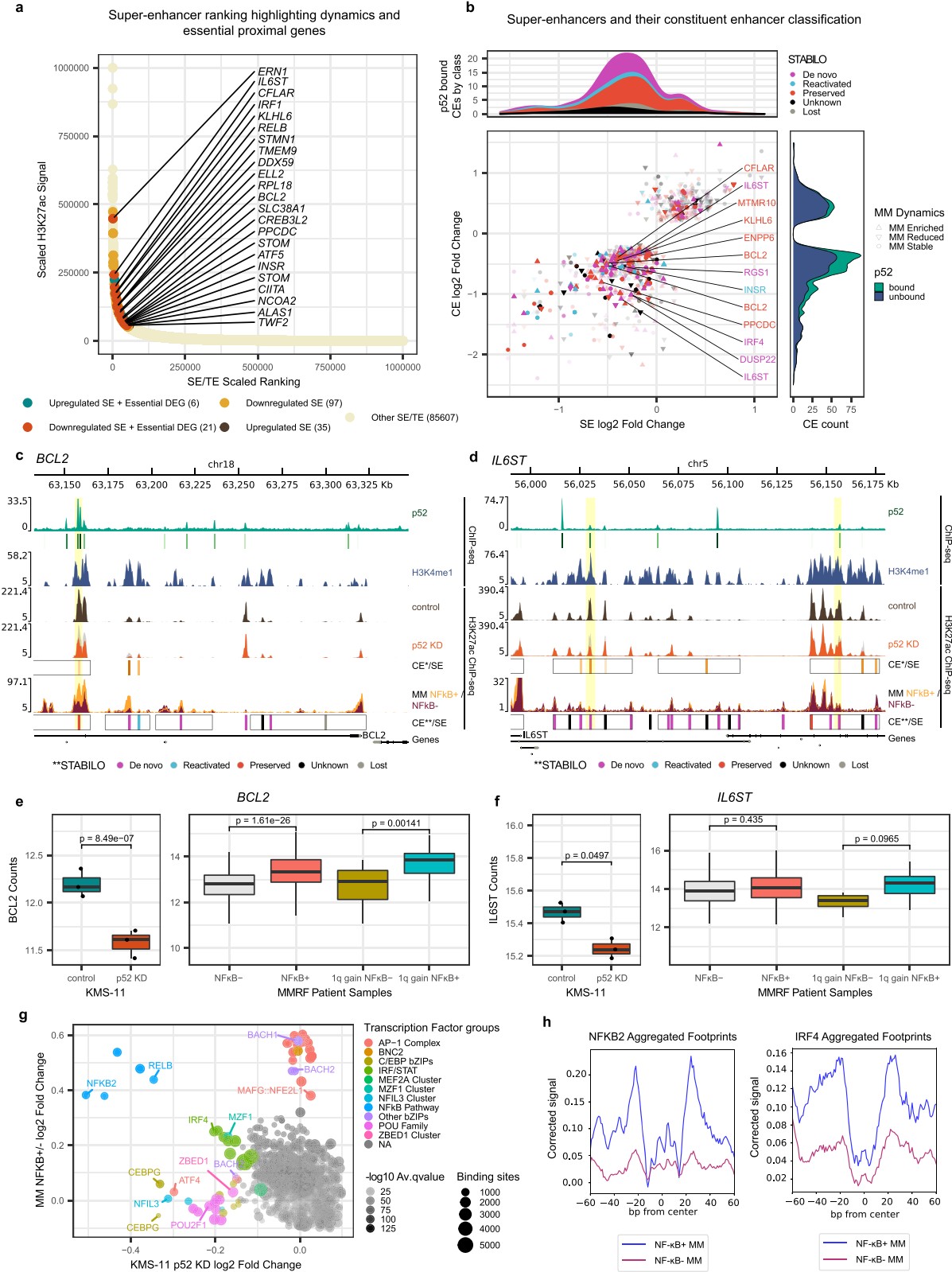

**a** Super-enhancer ranking highlighting dynamics and essential proximal genes

**b** Super-enhancers and their constituent enhancer classification

**c** BCL2    **d** IL6ST

**e** BCL2    **f** IL6ST

**g** **h** NFKB2 Aggregated Footprints — IRF4 Aggregated Footprints

Additionally, we were able to rescue all three deletion effects via reconstitution of RGS1 expression, consolidating the causal link between this de novo enhancer and the oncogenic functions of its target gene *RGS1* (Fig. 6g, h and Supp. Fig. 9j–l). Finally, we tested how deletion of this *cis*-regulatory element impacted tumour growth and survival as well as invasiveness in vivo using subcutaneous and orthotopic xenograft mouse models (Fig. 6I, j, Supp. Fig. 9m, n

and 10a). Deletion of the *RGS1* SE significantly decreased the tumour burden and prolonged survival in subcutaneous experiments while also reducing tumour nodules in the liver of mice subjected to the orthotopic experiments. Additionally, all these effects could be reversed by *RGS1* overexpression (Fig. 6I, j, Supp. Fig. 9m, n and 10a).

RGS proteins have been previously reported to regulate the JNK and p38α signalling pathways to mediate proliferative and apoptotic

**Fig. 4 | Dormant p52-dependent enhancers contribute to dynamic super-enhancers targeting essential myeloma transcriptional programmes.**
**a** Summarised ranking of SEs detected in KMS-11. SEs with significant signal dynamics and within proximity of essential genes undergoing similar dynamics are highlighted (<500 Kb). Signal and rank are scaled to 1 million and maximum signal values across replicates are plotted. **b** Super-enhancer versus constituent enhancer (CE) H3K27 acetylation dynamics following p52 knockdown in KMS-11. Opacity is reduced for CEs without p52 overlap. p52 binding status at CEs is summarised to the right with a kernel density plot. Enhancers overlapping p52, enriched in NF-κB+ samples (H3K27ac or accessibility relative to NF-κB- PB samples) and in proximity of essential genes (<500 Kb) are annotated. STABILO classification summarised atop with a kernel density plot for p52 bound enhancers. **c, d** Visualisation for *BCL2* and *IL6ST* loci encompassing proximal SEs. Tracks display: p52 ChIP-seq signal in green; H3K4me1 ChIP-seq signal in blue; ChIP-seq signal from control (brown) and p52 KD (orange) in KMS-11; SEs (black rectangles) and dynamic H3K27ac peaks (orange = lost) called in KMS-11; ChIP-seq signal from average of NF-κB+ (yellow) or NF-κB-

(dark blue) patients; SEs (black rectangles) and constituent H3K27ac peaks classified by STABILO; gene track. **e, f** *BCL2* and *IL6ST* expression in control (ctrl) and p52 knockdown in KMS-11 (*n* = 3 biological replicates per condition) as well as MMRF NF-κB+ (*n* = 331 patients) and NF-κB- (*n* = 298 patients) groups and 1q gain subtype subsets: 1q gain NF-κB- (*n* = 14 patients) and 1q gain NF-κB+ (*n* = 50 patients). Normalised rLog transformed counts. Lower and upper hinges correspond to first and third quartiles. Central value corresponds to the median. Whiskers extend to largest/smallest values no further than 1.5 x IQR (Interquartile range). *p*-values determined by 2-sided Wald test and adjusted for multiple comparisons (Benjamini-Hochberg) in DESeq2. **g** Transcription factor binding changes across SEs based on footprinting analyses in KMS-11 p52 KD and NF-κB+ ATAC-seq. Transcription factors in the top 5% of differential binding for each experiment are grouped by type/family/motif and coloured. **h** Aggregated footprints for NFKB2 and IRF4 transcription factors showing differential binding in NF-κB+ MM samples within the SEs identified in KMS-11.

signals[78]. We therefore evaluated the activation of signalling molecules involved in both the JNK and p38 activation cascade using WT, *RGS1* SE-del and RGS1 overexpressing MM cells (KMS-11 and LP1). Immunoblotting analyses showed RGS1 levels to be positively correlated with JNK/c-Jun pathway activation and negatively correlated with the p38/p53/Bax signalling axis (Fig. 6g and Supp. Fig. 9f). Additionally, ectopic expression of RGS1 resulted in GRAP2 upregulation, leading to the activation of JNK signalling and downstream pro-proliferative genes, *cyclin D1* and *cdc2* (Fig. 6g). Such effects were attenuated upon GRAP2 knockdown (Supp. Fig. 10b). These observations are consistent with previous reports demonstrating p38α mediated suppression of normal and cancer cell proliferation via inhibition of the GRAP2/JNK/c-Jun pathway[79,80]. Hence, it can be postulated that *RGS1* is a myeloma driver gene that exerts its oncogenic activity via suppressing p38α mediated antagonism of the GRAP2/JNK-cJun pathway (Fig. 6h).

Altogether, our data establish a significant role for p52 in reprogramming the myeloma epigenome. Particularly, p52 appears to be necessary for sustaining and potentially initiating the activity of MM specific regulatory elements that potentiate pro-oncogenic super-enhancer and target driver gene *cis*-regulatory pairings like *BCL2* and *RGS1*.

## Discussion

Aberrant non-canonical NF-κB signalling, driven by secondary mutations in regulators of the pathway, has been implicated as a critical step in the progression of MM[36–38]. Here, we defined the *cis*-regulatory landscape of NF-κB2/p52 during constitutive activation of non-canonical NF-κB in MM. We identified the multiple myeloma enhancers responding to elevated ncNF-κB activity, many of which are directly bound by p52 and either newly formed in MM or pre-established during normal B-cell development. Targeting the non-canonical NF-κB pathway through *NFKB2* knockdown resulted in the disruption of oncogenic transcriptional programmes shared with myeloma patients displaying high NF-κB activity (NF-κB+), along with the activity of p52-responsive enhancers. Notably, the altered transcriptomes included various genes that are critical for the survival and proliferation of myeloma plasma cells. We further revealed the significant role of NF-κB/p52 responsive enhancers and super-enhancer dynamics in activating such genes via *cis*-regulatory interactions. Taken together, our findings demonstrate that the sustained levels of ncNF-κB induction contribute to MM progression by activating a diverse repertoire of enhancers that diverges from normal B-cell maintenance. Our data suggest such p52-driven epigenome remodelling is at least partly responsible for facilitating the co-option of enhancers capable of supporting myeloma progression.

The extent of the p52-mediated epigenome changes observed are likely to be highly conditional on the constitutive mode of NF-κB activation in myeloma plasma cells that carry genetic dysregulation of

the pathway. Recent work investigating the temporal dynamics of NF-κB activation under different stimuli has highlighted that a sustained mode of NF-κB activation leads to much more profound epigenomic changes, that for instance, kickstart immune transcriptional programmes in macrophages[57]. Under this paradigm, the mutation-driven constitutive activation of NF-κB studied herein could be exploiting the sustained nature of non-canonical NF-κB[81] to mediate nucleosome displacement that could arguably exceed even natural inflammatory epigenome remodelling[82]. However, such changes cannot occur without the assistance of the chromatin remodelling machinery. Canonical NF-κB transcription factors such as p65 are already known to recruit and interact with transcriptional coactivators such as CBP/p300 or BRD4 to remodel the epigenome and form super-enhancers[56,83]. The comparably significant epigenomic dynamics involving p52 shown here are also likely to involve a number of important co-factors, especially those involved in mediating SE activity such as BRD4 or MED1[84]. To date, p52/RelB heterodimers have been reported to interact with SWI/SNF (SWItch/Sucrose Non-Fermentable) remodelling complexes via the requiem protein (REQ)[85]. More recently, p52 was also found to bind cooperatively with ETS1/2 or form part of a repression complex with HDAC4 to regulate transcription in cancers[86–88]. However, it remains unclear what other complexes are formed under constitutive activation in cancer genomes. Future investigation will be warranted to elucidate the major players involved in supporting the *cis*-regulatory roles of p52 relevant to myeloma.

Enhancer re-wiring is a phenomenon of increasing relevance in cancer[89]. Such events have recently been highlighted in multiple studies to support myeloma progression[21,22]. However, the transcription factors enabling these processes are currently uncharacterised. This work conclusively implicates NF-κB/p52 in reactivating B-cell and non-B-cell lineage enhancers. The primary functions of the non-B-cell lineage enhancers (de novo) appear to be incredibly diverse. It is feasible that the pleiotropic nature of NF-κB signalling in development and inflammation, has placed this pathway in a particularly exploitable position for oncogenic re-wiring. Once uncontrolled activation is gained, a diverse repertoire of enhancers is effectively unlocked for further selection and modulation during the clonal evolution of multiple myeloma[60]. Hyperactivation of canonical NF-κB has already been implicated in co-opting enhancers of lymphoid origin to drive metastasis in clear cell renal carcinoma through p65/p300 and HIF2A cooperativity near *CXCR4* gene[90]. Therefore, it is feasible that other translineage enhancers specifically targeted during inflammatory responses or other NF-κB dependent processes in other tissues are being exploited. Case in point, the p52-targeted enhancers reported interact with essential genes such as *BCL2* and *RGS1* to promote the survival of MM clones that underwent NF-κB-activating mutations. The *RGS1* enhancer in particular appears to represent a case of re-wiring. It involves an enhancer which is usually inactive in B-cells but can be

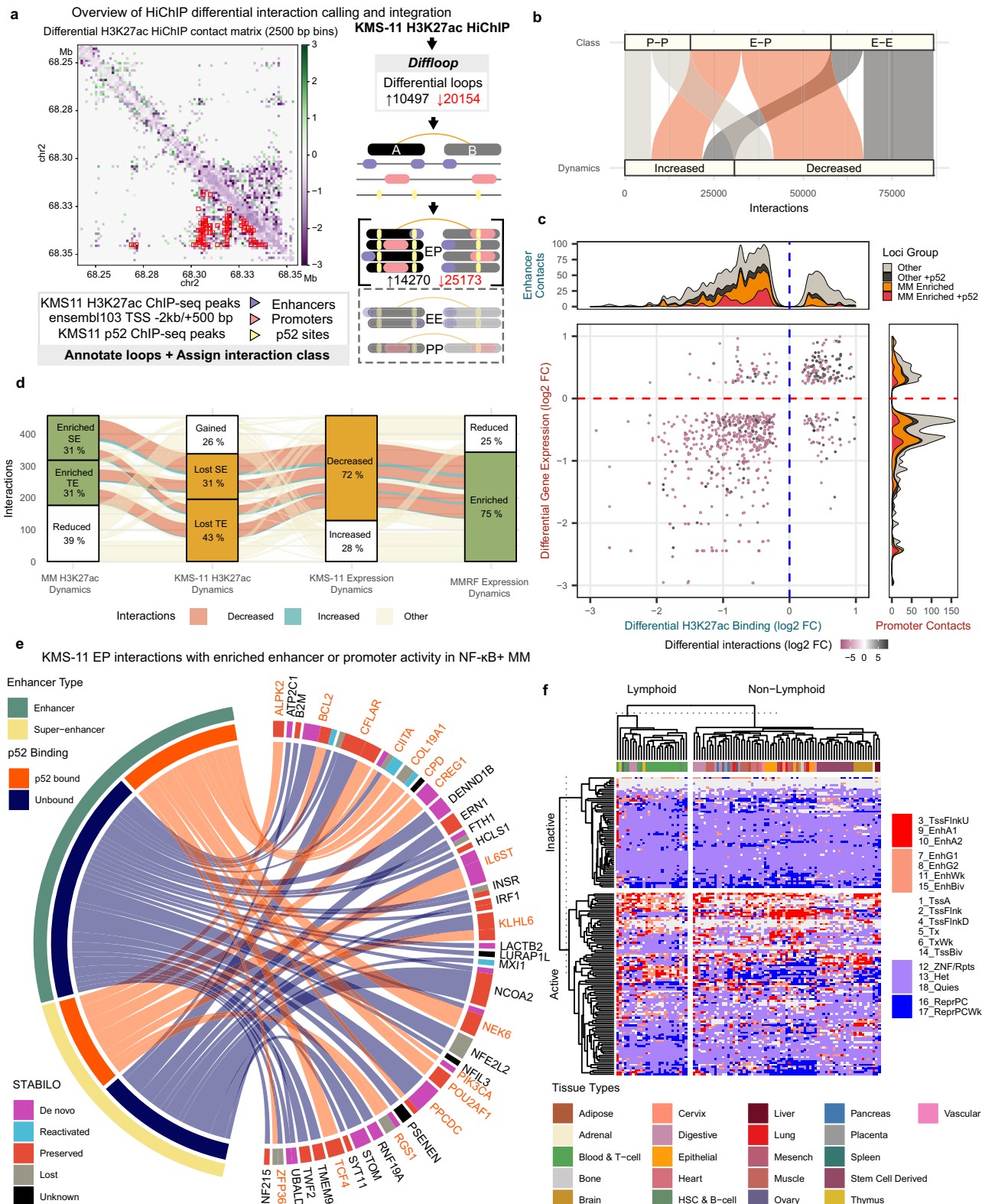

activated in a variety of other tissues including GM12878—known to undergo constitutive NF-κB signalling. Based on our results, this example of p52-mediated reprogramming is sufficient to disrupt the phosphorylation of p38, ultimately boosting JNK activity and impacting myeloma proliferation (Fig. 6h). Given pharmacological inhibitors for other GTPase family members have been identified[91], we are hopeful that this work will stimulate further research and the

development of specific RGS1 inhibitors that may contribute to viable therapeutic strategies to treat NF-κB+ patients.

Given the association between NF-κB+ and key subtypes exhibiting poor overall survival (MAF, 1q gain), it is possible subtype-specific epigenomic backgrounds may be predisposing towards p52-driven oncogenic re-wiring events and therefore lead to preferential accumulation of activating mutations such as *TRAF3* throughout the clonal

**Fig. 5 | Constitutive p52 remodels the myeloma interactome to reactivate dormant enhancers regulating key oncogenes. a** Overview of HiChIP significant loops and integration of differential loops with existing data. Both p52 KD and control loops are merged before differential analysis. Loop anchors are then annotated on the basis of existing H3K27ac peaks, RNA expression, p52 ChIP-seq and genomic features to assign loops to EE/PP/EP interactions before further integration with other data sources. The contact matrix displays the contact count fold change at the *PLEK* super enhancer cluster. **b** Classification of differential interactions detected. **c** H3K27ac and gene expression dynamics are paired on the basis of putative EP interactions they are associated with, effectively revealing a positive correlation. Bubble size combines false discovery rate estimates for enhancer and promoter dynamics. Colour encodes the differential interaction changes observed. Each enhancer or promoter feature is summarised on either side of the axes and classified depending on whether they are enriched in MM tumours and/or p52 bound. **d** Relationships between significant feature dynamics among EPI pairs with downregulated target genes resulting from p52 knockdown in KMS-11

cells. Most KMS-11 enhancers are downregulated and enriched in NF-κB+. Differential direction of the interactions is highlighted for connected features of interest only. **e** Chord diagram illustrating the characteristics and relationships of enhancers showing concordant downregulation of their H3K27 acetylation, interaction and target gene expression upon p52KD in KMS-11 cells and enriched epigenomic activity or transcription in NF-κB+ samples. Genes linked to a p52-bound enhancer are indicated in bold. **f** Activity at the MM-specific de novo enhancers identified as p52-bound and p52-dependent across a diverse collection of tissues. Active loci require at least 1 sample to show strong enhancer activity within the Non-Lymphoid sample group. The first column corresponding to sample GM12878 diverges from the rest showing high enhancer activity across the loci surveyed. The epigenomic state transitions across the Roadmap Epigenomics collection were summarised based on their 18-state chromHMM model segmentation. States 3,9 and 10 were most similar to the Strong Enhancer states defined for STABILO and are highlighted in red. Repressed or silent states were coloured blue and transcriptional states were coloured white.

evolution of MM tumours. Such a phenomenon could originate from the regulatory overlaps between complex programmes which simultaneously define subtypes but also arm p52-responsive enhancers with the wiring that ignites progression of MM upon p52-activation. Determining the interplay between subtype and NF-κB epigenomic programmes that leads to p52 mediated oncogenic re-wiring could be crucial to improving outcomes for such subtypes. Current findings indicate patients developing MAF or 1q gain subtype tumours are more likely to be affected by constitutive activation of non-canonical NF-κB and may especially benefit from additional screening and treatment options targeting NF-κB.

The increased activity of non-canonical NF-κB pathway and in particular, the transcription factor p52, has also been previously associated with other blood, pancreatic, skin, lung, colorectal and ovarian cancers[92–98]. Therefore, despite the central role of NF-κB signalling in B-cell malignancies, the *cis*-regulatory mechanisms uncovered during aberrant non-canonical NF-κB activation may not be strictly limited to multiple myeloma and could be exploited across other cancers or inflammatory disorders. Further characterisation of the gene targets, co-regulators and transcriptional complexes forming as a result of constitutive NF-κB/p52 activity will provide the foundation for new and more selective therapeutic strategies in fighting myeloma and other diseases that exploit the NF-κB pathway[99].

## Methods
This research complies with all relevant ethical regulations. Written, informed consent and ethical approval by the National Healthcare Group Domain Specific Review Board (NHG DSRB: 2007/00173) was obtained for human samples, in accordance with the Declaration of Helsinki. Animal studies were approved by Institutional Animal Care and Use Committee of Nanyang Technological University Singapore (NTU-ARF; AUP: A21070) and the Agency for Science, Technology and Research (A*STAR)—Institutional Animal Care and Use Committee (IACUC; protocol #221738). Tumour size/burden was monitored every 2–3 days. The maximal tumour volume approved by the IACUC was 2000 mm³. In some cases, this limit was exceeded on the last day of measurement and the mice were immediately euthanized.

### Statistics and reproducibility
Sex and gender were not considered in the study design as multiple myeloma progression has not been reported to be influenced by either. Data derived from patients featured both sexes ($n = 7$; 5 male, 2 female). Data derived from animals featured both sexes ($n = 30$; 21 male, 9 female). No statistical method was used to predetermine sample size. No data were excluded from the analyses. Only animal experiments were randomized. The investigators were not blinded to allocation during experiments and outcome assessment.

### Cell culture
Human multiple myeloma cell lines, including MM1.144, L363, ANBL6, U266, KMS-11, RPMI8226, JJN3, XG7 and H929, were gifts from Prof. Leif Bergsagel (Mayo Clinic, Scottsdale, AZ, USA), and were authenticated by short tandem repeat (STR) profiling. The MM1.S cell line was obtained from ATCC while LP1 and MOLP8 were obtained from the German Collection of Microorganisms and Cell Cultures. All cell lines were tested to be mycoplasma-free using Mycoplasma PCR Detection Kit (Abm, G238) prior to experiments. MM cell lines were cultured in RPMI 1640 media with 2.05 mM L-Glutamine (SH30027.01; Hyclone). XG7 and ANBL6 were supplemented with 2 nM IL-6 (PeproTech) while H929 was supplemented with 0.05 mM β-mercaptoethanol (Sigma). 293T cells were cultured in Dulbecco's modified Eagle's medium (DMEM) with 4 mM glucose (high glucose) (SH30243.01; Hyclone) and passaged with 0.25% Trypsin/EDTA (Gibco). All cell cultures were supplemented with 10% Foetal Bovine Serum (Sigma). Cultures were incubated at 37 °C and 5% $CO_2$.

### Western blotting
Cell lysates were extracted using ice-cold RIPA lysis buffer (10 mM Tris-HCl, pH 8.0, 1 mM EDTA, 0.5 mM EGTA, 1% Triton X-100, 0.1% Sodium Deoxycholate, 0.1% SDS, 140 mM NaCl, 0.5 mM DTT, 0.3 mM NaVO3) with 1 x protease inhibitor cocktail (Merck). 50 μg of total protein was resolved by SDS-PAGE under reducing conditions and transferred to 0.2 μm polyvinylidene difluoride (PVDF) membrane in Tris-glycine buffer with 10% methanol. Membranes were individually probed with NFKB1(13586; CST, 1:1000), NFKB2 (3017; CST, 1:1000), RelB (10544; CST, 1:1000), p65 (8242; CST, 1:1000), NIK (4994; CST, 1:1000), TRAF3 (61095; CST, 1:1000), RGS1 (ab154973; abcam, 1:1000), pJNK (4668S; CST, 1:1000), p-cdc2 (4539S; CST, 1:1000), cyclin D1 (E3P5S) (55506; CST, 1:1000) and p-p38 (4511S; CST, 1:1000) rabbit monoclonal antibody. GAPDH (sc-32233; Santa Cruz, 1:10000), c-Rel (sc-6955; Santa Cruz, 1:500), p-IKBa (9246; CST, 1:1000),BCL2 (15071, CST, 1:1000), JNK (sc-7345; Santa Cruz, 1:500), c-Jun (sc-74543; Santa Cruz, 1:500), p-c-Jun (sc-822; Santa Cruz, 1:500), p38α (sc-166182; Santa Cruz, 1:500), p53 (2524S; CST, 1:1000), p-p53 (sc-377567; Santa Cruz, 1:500), GRAP2 (sc-73652; Santa Cruz, 1:500), cdc2 (sc-53219; Santa Cruz, 1:500) and Bax (sc-7480; Santa Cruz, 1:500) mouse monoclonal antibody. Binding was detected by anti-rabbit (7074S; CST, 1:10000) or anti-mouse (sc-516102; Santa Cruz, 1:10000) horseradish peroxidase (HRP)-conjugated secondary antibody. Bands were visualized with Clarity Max ECL Substrate (1705062;BioRad). Images were taken with Chemidoc XRS imaging system (BioRad) and quantified using GelQuant.NET software provided by biochemlabsolutions.com (CA, USA).

### CoMMpass data and NF-κB indexing
Gene counts and mutation data were obtained from CoMMpass release IA14a. DESeq2 v1.30.0 was used to normalise and transform

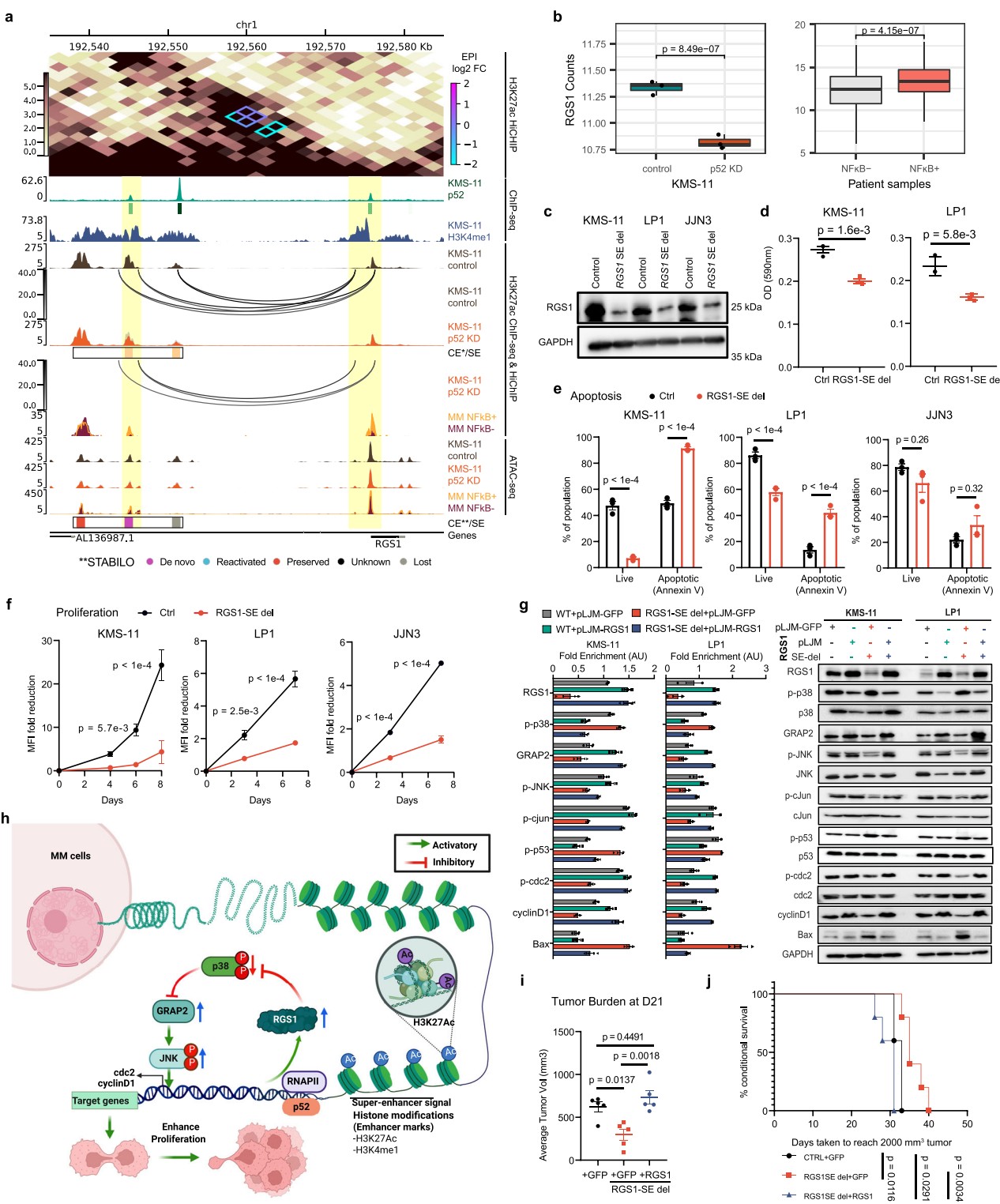

counts (VST)[100]. The NF-κB index was calculated as the geometric mean of 11 genes *BIRC3*, *NFKB2*, *TNFAIP3*, *IL2RG*, *NFKBIE*, *RELB*, *NFKBIA*, *CD74*, *PLEK*, *MALT1* and *WNT10A*[37]. The values were converted to a standard score to create the index. Samples with a positive index and at least one gene displaying expression beyond its upper quartile were classified NF-κB+ and the remaining as NF-κB-. Samples were categorised into putative subtypes on the basis of previous RNA subtyping work performed on the CoMMpass data[18]. Associations between NF-κB groups and subtypes were evaluated by Fisher's exact test.

**Endogenous p52, H3K27ac, H3K4me1 and H3K27me3 ChIP-seq**

Cells were harvested and resuspended to a density of $1 \times 10^6$ per ml of culture media. 37% formaldehyde was added to a final concentration of 1% for 12 min then quenched with 0.125 M Glycine. The cells were then swelled with SDS buffer (0.1 M NaCl), 0.05 M Tris-Cl pH 8, 0.005 M EDTA pH 8, 0.5% SDS, 1 x protease inhibitor cocktail then lysed in IP buffer (SDS buffer + Triton buffer (0.1 M NaCl, 0.1 M TrisCl pH 8, 0.005 M EDTA pH8, 5% Triton X-100, 1 x protease inhibitor cocktail) in a 2:1 ratio) for nuclei isolation. Chromatin samples were sheared using

**Fig. 6 | RGS1 is a target of a p52 dependent *cis*-acting super-enhancer that drives myeloma progression via JNK and p38α signalling. a** Visualisation for the *RGS1* locus encompassing proximal SEs. Tracks display: H3K27ac HiChIP contact matrix with differential loops annotated and coloured by log2 fold change; p52 ChIP-seq signal; ChIP-seq signal from control (brown) and p52 KD (orange) in KMS-11; SEs and dynamic H3K27ac peaks (orange = lost) called from KMS-11; loops indicated by arc annotation on control and p52 KD ChIP-seq tracks and coloured by counts per million; ChIP-seq signal from average of NF-κB+ (yellow) or NF-κB- (dark red) patients; equivalent ATAC-seq signals from KMS-11 and patients; SEs and constituent H3K27ac peaks classified by STABILO; gene track. **b** *RGS1* expression in control and p52 knockdown in KMS-11 (left, *n* = 3 biological replicates per group) as well as MMRF groups (right): NF-κB+ (*n* = 331) and NF-κB- (*n* = 298). Normalised rLog transformed counts. Lower and upper hinges correspond to first and third quartiles. Central value corresponds to median. Whiskers extend to largest/smallest values no further than 1.5 x IQR (Interquartile range). 2-sided Wald test, adjusted for multiple comparisons (Benjamini-Hochberg) in DESeq2.

**c** Representative western blot of RGS1 protein expression upon SE deletion, *n* = 2. **d** Cell adhesion to fibronectin. Points represent mean OD at 590 nm of technical triplicates. Mean ± SEM of 3 biological replicates per condition. Two-tailed *t*-test. **e** Assesment of apoptosis by FACS. Percentage of live cells (Annexin V-) and apoptotic cells (Annexin V + ) shown. Mean ± SEM of 3 biological replicates per condition. 2-way ANOVA. **f** Proliferation analyses by dye dilution. MFI (mean fluorescence intensity) calculated relative to day 0. Mean ± SEM of 3 biological replicates per condition. 2-way ANOVA. **g** Representative western blot showing activation level of JNK/c-Jun and p38α/p53/Bax axis of signalling cascade. Bands normalized with respect to GAPDH and displayed relative to WT+pLJM-GFP. Mean ± SEM of *n* = 3 biological replicates per condition. **h** Graphical summary of proposed p52/RGS1/JNK regulatory mechanism impacting MM. Created with BioRender.com **i** Mean tumour volume at D21 ± SEM, 5 mice per group. One way ANOVA. **j** Kaplan–Meier survival curves of mice xenografts. Survival evaluated from first day until sacrifice. *N* = 5 mice for each group, two-sided log-rank test. Source data are provided as a Source Data file.

a sonicator set at high with 30 s pulse on/off cycles (Diagenode Bioruptor). Pulse cycles were optimised (9–15 cycles) to achieve complete shearing of samples while keeping the input smear of the sample between 100 – 500 bp when run on agarose gel. After pre-clearing with Sepharose G beads, 1% of chromatin was taken for input while the remaining was split equally before adding one of the following antibodies: NFKB2 antibody (A300-BL7039; Bethyl Laboratories), H3K27ac antibody (07–360; Merck), H3K27me3 (9733S, CST), H3K4me1 (AB8895-1003, Abcam) or IgG Rabbit (P120-101; Bethyl Laboratories). Elution of input and ChIP material was done by incubating beads in elution buffer (1% SDS, 0.84% NaHCO3, 0.25 M NaCl) overnight at 65 °C. Additional data for non-mutant MMCLs (H929 and KMS-28BM) was retrieved from GSM4338494, GSM4338495, GSM4338486 and GSM4338487[20].

### Processing and RNA sequencing of NUHS patient samples
Bone marrow (BM) samples derived from 10 patients diagnosed with MM were previously collected as part of development of a comprehensive tissue and gene registry for haematological malignancies by the National University Health System (NUHS). Written, informed consent and ethical approval by the institutional review board (IRB) was obtained (NHG DSRB: 2007/00173), in accordance with the Declaration of Helsinki. Detailed clinical characteristics were obtained from patients' health records and previously published in supplemental table 1 in[20]. CD138 positive plasma cells were purified from available matching BM aspirates using immunomagnetic bead selection (EasySep™; Stem Cell Technologies). Total RNA from 7 of these patients was extracted from CD138+ cells using the miRNeasy Mini Kit (Qiagen GmbH, Germany) according to the manufacturer's instruction. Ribosomal RNA was removed with NEB Ribo-Zero Plus rRNA Depletion Kit (New England Biolabs, Cat No. 20037135) prior to RNA-seq library preparation. The RNA library construction and RNA-sequencing service were provided by Novogene Singapore. To prevent any potential batch effects, the libraries were sequenced on Illumina HiSeq4000 lanes (150 bp, paired-end seq,110 G raw data per lane) in the same flowcell. The RNA-seq data were subsequently processed together with other data sources detailed in the next section. Raw and processed files were deposited in the GEO database under GSE230296.

### Processing of existing patient RNA-seq, ChIP-seq and ATAC-seq
Patient RNA-seq reads were trimmed using trimmomatic[101], mapped to GRCh38 using hisat2 v2.1.0[102] and counted with featureCounts (subread v2.0.0)[103]. Read counts were input into DESeq2 for normalisation and rLog transformation prior to dimensionality reduction (PCA)[100]. The transformed counts were also used for the NF-κB index calculation as previously detailed to group matching ChIP and ATAC-seq patient samples. CoMMpass differential expression results are used for subsequent integrations.

Somatic mutations were putatively called from the RNA-seq reads using GATK4[104]. Briefly, the reads were pre-processed through the following recommended steps: Mapping, Mark Duplicates, SplitNCigarReads and Base Recalibration. Somatic variants were then called using Mutect2 and filtered with recommended parameters to account for contamination, orientation bias and germline variants. Finally variants were annotated with Funcotator and visualised in maftools[105].

Raw patient ChIP/RNA/ATAC-seq reads were downloaded from PRJEB25605[19]. ChIP-seq reads were mapped to GRCh38 using bowtie2 v2.3.5.1[106]. Reads with MAPQ < 10 were dropped and the resulting alignments used for calling broad peaks with MACS2 v2.2.7.1[107]. Peaks overlapping known blacklisted regions were dropped[108]. Additionally, pre-processed (GRCh38) ChIP-seq data matching the NUH patient RNA-seq was obtained from PRJNA608681[19].

Patient ATAC-seq reads were processed through the ENCODE ATAC-seq pipeline (https://github.com/ENCODE-DCC/atac-seq-pipeline) with mapping against GRCh38 with bowtie2[106]. Fragment counts within features of interest (p52 or H3K27ac peaks) were obtained with featureCounts in subread v2.0.0[103]. Both ChIP-seq and ATAC-seq counts were initially tested for differential expression in DESeq2 contrasting NF-κB+ vs NF-κB- (MM + PB) samples. Subsequent integrations contrast NF-κB+ vs NF-κB- (PB only) samples to maximise sensitivity[100]. BAMscale and deeptools were used to generate coverage bigwigs from the alignments across regions of interest for visualisation[109,110].

### Integration with ChromHMM state transitions across B-Cells (STABILO)
ChromHMM segmentation files generated in[21,22] were downloaded from the EBI FTP portal (BLUEPRINT release 20160816) and the Ordonez data portal (http://resources.idibaps.org/paper/chromatin-activation-as-a-unifying-principle-underlying-pathogenic-mechanisms-in-multiple-myeloma) respectively[21,22]. Ordoñez and colleagues assigned 12 ChromHMM states across 3 thymus derived naïve B-Cell, 3 blood derived naïve B-Cell, 3 germinal centre derived B-Cell, 1 non-class switch memory B-Cell, 2 class-switched memory B-Cell, 3 thymus derived plasma cell and 4 multiple myeloma samples. Alvarez-Benayas and colleagues assigned 12 ChromHMM states across 6 naïve B-Cell, 2 germinal centre derived B-Cell, 1 non-class switch memory B-Cell, 2 class-switched memory B-Cell, 2 plasma cell, 1 cell line (U266) and 4 multiple myeloma samples. We made a conservative selection of emission states (E2 and E6 from the Ordonez predictions and E9 and E10 from the Alvarez-Benayas predictions) to define active enhancers[111]. Such states were consistently marked by high H3K27ac and high H3K4me1. To summarise chromatin state transitions between inactive and active enhancers from B-Cell to plasma cell to myeloma at loci of interest in our study (p52 binding or H3K27ac sites) we first intersected the chromHMM states with the target regions using bedtools requiring 50% overlap[112]. Loci with active/inactive enhancer state

overlaps were summarised across B-Cells, plasma cells and myeloma samples. For the STABILO classification we labelled inactive loci gaining activity only in myeloma samples as De novo; loci active in B-cells but inactive in plasma cell and then re-activated in myeloma as Re-activated, loci active throughout samples as Preserved, loci active in B-cell or plasma cell samples but inactivated in myeloma as Lost; loci inactive throughout as Unknown.

## CRISPR p52 knock down

Cloning of lentiviral plasmidsFor KMS11, gRNAs were designed using the Broad Institute GPP sgRNA Designer/CRISPick[113] to target exon 6 of *NFKB2*. They were then cloned into TLCV2 (Addgene plasmid # 87360, a gift from Adam Karpfand) as previously described[114,115]. The following oligos were used to generate the TLCV2-NFKB2-Ex6 plasmid: NFKB2Ex6F: CACCTCCTAGATCTGTAACTACGA, NFKB2Ex6R: AAATC GTAGTTACAGATCTAGGAA. The plasmids were transformed into competent Stbl3™ bacteria via heat shock at 42 °C and plated on LB agar plates containing Ampicillin which were incubated overnight at 37 °C. Cycle sequencing (BigDye Terminator v3.1; 1st Base) was conducted to verify the success of cloning.

For *NFKB2* knock down in MM.1.144, sgRNA targeting exon 6 of *NFKB2* was cloned into lentiCRISPR v2 (Addgene plasmid # 52961), which is a constitutive CRISPR system. The guide sequence and cloning protocol followed are similar to that used for KMS11.

**Lentivirus production.** Two million 293T cells were seeded in a poly-L-lysine-coated 100 mm dish 1 day prior to transfection. Using a four-plasmid system, 6.5 μg pMDL, 3.6 μg VSV.G, 2.5 μg Rev and 10 μg TLCV2 was added to 62.5 μl of 2.5 M CaCl2 and topped up to 500 μl with distilled H2O. The sample was then added dropwise to 500 μl of 2x HEPES buffered saline adjusted to pH 7.07 (280 mM NaCl, 50 mM HEPES, 1.5 mM Na2HPO4, 10 mM KCl, 12 mM Dextrose) and left to incubate for 20 min at room temperature before adding dropwise to the cells. After 6 h, the cells were washed with PBS and fresh DMEM was added and left overnight. The media was then collected and passed through a 0.45 μm filter before aliquoting and storing at -80 °C.

A spin infection was used to transduce KMS-11 cells with TLCV2-containing virus adapted from[116]. In brief, 5 x 10⁵ KMS-11 cells were resuspended in 1 mL of lentivirus containing either TLCV2 or TLCV2-Ex6. Polybrene was added to a final concentration of 8 μg/ml. The cells were centrifuged for 45 min at 400 g at room temperature. The cells were then gently resuspended and plated in a 6 well plate. The next day, the virus was removed, and fresh media added. After another 24 h, cells were selected using puromycin added at a concentration of 0.5 μg/ml for 72 h.

To induce the expression of Cas9 and mediate the knockdown of *NKFB2*, cells were treated with 1 μg/ml Doxycycline before sorting for GFP-positive cells (Aria 3 Sorter; BD).

For lentiCRISPR v2 system, post viral transduction, cells were selected under puromycin pressure and harvested for H3K27ac ChIP-seq, RNA-seq and ATAC-seq on day 10 post transduction.

## Generation of cell line RNA-seq, ChIP-seq, ATAC-seq and H3K27ac HiChIP

For RNA-seq, RNA was isolated from TLCV2-Ex6 transduced KMS-11 and MM1.144 cells using phenol-chloroform extraction and treatment with TURBO DNAfree kit (Ambion), followed by rRNA removal using Ribo-Zero (Illumina) according to manufacturers' recommendations. Total cDNA libraries were prepared using NEBNext® Ultra™ II Directional RNA Library Prep Kit (NEB #E7765).

ChIP libraries were prepared using NEBNext Ultra II DNA Library Prep Kit for Illumina (NEB) and underwent paired end sequencing in a NextSeq500 or HiSeq PE150 (Novogene).

ATAC-seq was performed on KMS-11 cells transduced with TLCV2-Ex6 as previously described[117,118]. Briefly, 50,000 cells were washed with cold PBS at 500 × g at 4 °C for 5 min. The cells were resuspended in 50 μL of cold ATAC-Resuspension Buffer (RSB) (10 mM Tris-HCl, pH 7.4, 10 mM NaCl, 3 mM MgCl2) containing 0.1% NP40, 0.1% Tween20, and 0.01% Digitonin (Promega #G9441) and incubated on ice for 3 min. The cells were then washed in RBS containing 0.1% Tween-20 at 500 g at 4 °C for 10 min. The nuclei were subjected to the transposase reaction for 30 min at 37 °C; termination of the reaction and DNA purification was performed using Zymo DNA clean and Concentrator kit (cat # D4014). The purified DNA was amplified as described before with NEBNext High-Fidelity 2x PCR Master Mix (New England Biolabs). The PCR amplified product was purified and size selected twice with AMPure beads (Beckman).

The library profile and quality was verified using 2100 Bioanalyzer (Agilent Technologies) and the libraries were quantified using Qubit kit (Thermofisher). Subsequent paired-end sequencing of cDNA libraries for RNA-Seq and ATAC-Seq were performed using HiSeq PE150 sequencing platform (Novogene).

For H3K27ac Hi-ChIP, 20 x 10⁶ TLCV2-Ex6 transduced KMS-11 cells were collected and washed once with cold PBS. They were then aliquoted into 4 tubes of 5 x 10⁶ cells and the cell pellets were snapped frozen in liquid nitrogen before shipping to be processed by Dovetail®.

## Processing of cell line p52 and H3K27ac ChIP-seq data

MM1.144 ChIP-seq reads were processed through the ENCODE ChIP-seq pipeline (https://github.com/ENCODE-DCC/chip-seq-pipeline2) with mapping against GRCh38 with bowtie2[106]. Fragment counts within features of interest (p52 or H3K27ac peaks) were obtained with featureCounts in subread v2.0.1[103]. For knock down experiments ChIP-seq counts were tested for differential expression in DESeq2. BAMscale and deeptools were used to generate coverage bigwigs from the alignments across regions of interest for visualisation[109,110].

Remaining ChIP-seq datasets were aligned against hg38 using bwa mem[119], with duplicate reads marked using samblaster[120]. The resulting alignments were filtered to remove duplicate reads, poorly mapping reads and those found to originate from ENCODE blacklisted region sites[108]. Peaks were identified using MACS2[107] and dataset quality was confirmed using ChIPQC[121]. The resulting peaks were processed using DiffBind[122], and resized to 500 bp, centred on the middle of the peak in order to identify consensus peaksets across all of the cell types investigated. Differential binding analysis of H3K27ac profiles between WT and p52 KD KMS-11 was performed using DiffBind using a window size of 1 Kb.

All peaks were annotated as overlapping intronic, intergenic, exonic or promoter regions using GENCODE version 35. Genomic distribution of all peaks was evaluated across the GENCODE v35 annotation via annotatePeaks.pl (HOMER v4.11) and plotted with ggplot2[123,124].

De novo motif identification was performed on endogenous peak sets using MEME[125], which identified a motif (Supplementary Fig. 2b) closely resembling the NFKB2 (p52) motif present in JASPAR[126]. In addition, these peaks were scanned FIMO[127] to identify the location of the highest scoring motifs with respect to the peak centre to create TFBS-landscape plots (Supp. Fig. 2c, d)[128]. Differential binding analysis of H3K27ac profiles between mutant MMCLs and non-mutant MMCLs was performed using Deseq2. Differential binding analysis of H3K27ac profiles between WT and p52KD KMS-11 was performed using DiffBind using a window size of 1 Kb.

## Processing of p52 knock down cell line RNA/ATAC-seq data

RNA-seq reads obtained from KMS-11 and MM1.144 knock down experiments (treated vs control) were trimmed using trimmomatic and mapped to GRCh38 using STAR (on Galaxy Australia) and hisat2 v2.1.0 respectively[102,129,130]. Fragment counts were obtained using featureCounts (Subread v 2.0.1) against GENCODE v35 and input into DESeq2 v1.30.0 running on R v4.0.3 to perform the standard

differential testing workflow[100,131]. Biological replicate batches were included as co-variate in the statistical model and batch correction of rLog transformed counts performed with limma v3.46.0[132]. FDR was estimated using the default Benjamini-Hochberg procedure and via independent hypothesis weighting (IHW v1.18)[133].

ATAC-seq reads were processed through the ENCODE ATAC-seq pipeline and converted to bigwigs for visualisation as detailed for the patient ATAC-seq. The peaks output from the pipeline were merged by condition and then further merged on basis of 50% reciprocal overlap using bedtools to assign common or divergent consensus peaks[112]. The counts were subsequently also tested for differential expression in DESeq2 with batch correction[100].

### Processing of KMS-11 p52 knock down Hi-ChIP data

HiChIP reads were processed following the Dovetail fastq-to-valid-pairs pipeline (https://hichip.readthedocs.io/). Briefly, reads were mapped to GRCh38 using bwa mem with the following parameters: -5SP -T0[119]. The alignments were then parsed, sorted, deduplicated and split with pairtools (https://github.com/open2c/pairtools). The resulting valid pairs files were converted to contact matrices using cooler[134]. For TAD calling, 50KB normalised and corrected (KR) contact matrices were input into hicFindTADs with delta and qval parameters of 0.01 followed by hicDifferentialTAD. Contact matrices displayed in genome browser tracks were normalised and transformed to observed/expected counts using HiCExplorer[135]. Significant interactions as called by FitHiChIP were used for aggregate peak analyses. FitHiChIP was run in L + M mode on each condition with a bin size of 2500 bp, lower/upper distance thresholds of 10KB/2MB and coverage bias regression[136]. The resulting significant interactions (FDR ≥ 0.1) were parsed and input into DiffLoop for differential testing on the basis of contact count differences (via edgeR)[137,138]. Normalised contact matrices centred on the PLEK locus were compared for control and p52 KD conditions for illustration in Fig. 5a.

### Integrative superenhancer analysis

Superenhancers were called sample-wise from broad peaks obtained in KMS-11 and patient ChIP-seq data using a Python 3 implementation of ROSE with 12500 bp stitching distance and 2500 bp exclusion from TSS[84,139]. Super-enhancers were merged using bedtools to form consensus SEs and re-assigned to individual samples on basis of their overlap with the original SE coordinates for each sample (≥50%). Consensus enhancer peaks were assigned as constituent enhancers for each consensus super-enhancer forming the basis for fragment counting and differential testing using featureCounts and Deseq2 respectively[100,103].

### Integrative HiChIP analysis

Differential loop anchors spanning 2500 bp were annotated on the basis of their overlap with differential H3K27ac peaks and differential promoters (2 kb upstream and 500 bp downstream of all GENCODE v35 genes differentially expressed) identified in KMS-11 using bedtools and compiled into a master table in R[112,131]. Ambiguous loops with anchors overlapping multiple features were resolved using an inclusive strategy preserving all possible interactions. All interactions were then classified (PP, EP or EE) and filtered to focus on the abundant putative EPIs. Additional data (STABILO classification, p52 binding, SE, patient epigenomic and expression dynamics, DepMap information) linked to the interacting features were then integrated into a master table for further filtering and plotting in R using a range of visualisation packages[124,131,140–142].

### Luciferase assay in p52-overexpressing 293T cells

The ORF of *NFKB2* was cloned into pLJM.EGFP lentiviral vector using AgeI and EcoRI restriction sites followed by overexpression in 293T cells. The primers used for cloning are: p52-pLJM1-AgeI: AAA ACCGGT ATGGAGAGTTGCTACAACCCA, p52-pLJM1-EcoRI AAA GAATTC TCACGCCAGGCCGAACAG. 293T cells were transduced using a spin infection with virus containing the pLJM-p52 and pLJM-GFP (vector control) plasmids using a protocol adapted from[116] before selection with 1 µg/ml puromycin.

Luciferase reporter with minimal promoter (pGL4.23, Promega) and *Renilla* luciferase plasmid (pGL4.74, Promega) were co-transfected into 293T cells. KMS-11 gDNA was used as a template to amplify the constituent enhancer of each superenhancer region of interest using PCR and then inserted into pGL4.23 vector using XhoI and HindIII or KpnI. The primers used are: BCL2SE_XhoI: CCGCTCGAGTTTTCC AAAATGGTACCCTGAG, BCL2SE_HindIII: CCCAAGCTTAAGGGAAATC AACAGCACGT, RGS1SE_KpnI: GGGGTACCGATTACAGGCATGAGCCA, RGS1SE_XhoI: CCGCTCGAGCACCCAGCTAATTTTTGTATTT, IL6ST_ KpnI: GGGGTACCGTGGTACTTTGTTATGGCAG, IL6ST_XhoI: CCGCT CGAGTGAGAAATCTCTGCATTGTTTT. The luciferase assays were carried out with Dual-Luciferase Reporter Assay System (E1910, Promega).

### Deletions of regulatory elements

For super enhancer deletion, we designed two different gRNAs targeting a section of the super enhancer found to loop to the target gene promoter. They were cloned separately into lentiCRISPR v2 (Addgene plasmid # 52961, a gift from Feng Zhang) and lentiGuide-Hygro-dTomato (Addgene plasmid #99376, a gift from Kristen Brennand) as previously described[114]. Oligos used to generate the plasmids are listed in Supplementary Table 1. Cells were transduced using a spin infection with virus containing the relevant plasmids using a protocol adapted from[116]. In brief, cells were resuspended in lentivirus containing the relevant plasmids and polybrene was added to a final concentration of 8 µg/ml. The cells were centrifuged for 45 min at 400 g at room temperature and incubated at 37 °C and 5% CO2 overnight. Cells were first transduced with lentiGuide-Hygro-dTomato and selected with 300 ng/ml hygromycin for 10 days. The cells containing lentiGuide-Hygro-dTomato plasmids were then transduced with the lentiCRISPR v2 plasmid and selected with 1 µg/ml puromycin for 72 h.

To assess super enhancer deletion efficiency of the pool of sorted cells, genomic DNA was extracted and PCR was performed using primers flanking the genomic target region. A shorter PCR product is expected for deleted alleles compared to the uncut control. The PCR product for the shorter band was sequenced to verify specificity of the band and confirm the deletion of the region. Primers used are listed in Supplementary Table 1.

### Overexpression rescue of targets

To confirm the functional role of RGS1, rescue experiment was performed using RGS1 overexpression clone. The ORF of *RGS1* was cloned into pLJM.EGFP lentiviral vector (Addgene#91980) using NheI and AgeI restriction sites followed by overexpression in both control and SE deleted cells. The primers used for cloning are provided in Supplementary Table. 1. Cells were transduced using a spin infection with virus containing the pLJM-RGS1 and pLJM-GFP (vector control) plasmids using a protocol adapted from[116]. The experiment was setup in four conditions: WT cells + pLJM-GFP, WT cells + pLJM-RGS1, RGS1 SE-del + pLJM-GFP and RGS1 SE-del + pLJM-RGS1. Cells were transduced on Day3 post puromycin selection following SE deletion and incubated with the virus for 24 h. Following incubation, virus was removed, and cells were allowed to recover for 3–4 days before harvesting for western blotting and functional assays. To decipher the link between RGS1 and GRAP2 activation, shRNA mediated knock down of GRAP2 was performed in RGS1 over-expressing MM cells. The primers used for shRNA cloning are provided in Supplementary Table 1. Knock down cells were further selected under puromycin pressure of 0.5 µg/ml for 48 h followed by recovery of 48 h and harvested for western blotting.

## Tumour xenograft assay

Balb/c RAG -/- IL2Rγ -/- mice (6–10 weeks) were subcutaneously injected with $5 \times 10^6$ cells into the right hind flank. Five mice were used in each condition. The conditions tested were CTRL + GFP, RGS1SE-del +GFP and RGS1SE-del+RGS1. Mice were weighed and tumours measured every 2–3 days. For the survival analysis, mice were determined to have reached humane endpoint and sacrificed when tumour volume (calculated as $\frac{1}{2}(length \times width^2)$) reached the maximal tumour volume permitted by our IACUC ($2000 \text{ mm}^3$). Upon reaching endpoint, mice were euthanised via carbon dioxide overdose followed by cervical dislocation. In some cases, the maximal tumour volume was exceeded on the last day of measurement and the mice were immediately euthanized. Animal studies were approved by Institutional Animal Care and Use Committee of Nanyang Technological University Singapore (NTU-ARF; AUP: A21070). The housing condition for mice was temperature of 21–25 °C, relative humidity (RH) of 55–60% and pressure (Pa) of 5–8.

## In vivo orthotopic engraftment of tumour cells

Animal experiments were conducted according to the A*STAR approved IACUC protocol (#221738). NOD-*scid IL2ry^null* (NSG) mice were procured from The Jackson Laboratory (stock #005557), bred, and kept under conditions free of specific pathogens with a 12 h light-dark cycle with a temperature between 21–23 °C and humidity between 55–70%. NSG mice were allocated randomly into groups of 5. Each group was engrafted with either $2 \times 10^6$ of *RGS1* SE deleted KMS-11 (CRISPR), *RGS1* SE deleted + RGS1 overexpression KMS-11 (rescue), or KMS-11 (control) cells via tail vein injection. They were monitored with endpoints comprising of death or euthanasia ($CO_2$ chamber) on humane grounds (specifically hind limb paralysis). Livers were isolated post-mortem.

## Proliferation and viability assays

For cell proliferation, $5 \times 10^5$ KMS-11 cells were labelled with 5 μM CellTrace™ Blue dye (Invitrogen) for 20 min at 37 °C in the dark. Cells were sampled at multiple timepoints as indicated and analysed by flow cytometry (BD LSRFortessa™ X-20) to track loss of dye due to proliferation. MFI (mean fluorescence intensity) fold reduction relative to day 0 was subsequently calculated.

Cell viability was measured using the Alexa Fluor™ 488 Annexin V conjugate with Propidium Iodide (ThermoFisher Scientific) according to the manufacturer's instructions and analysed by flow cytometry (BD LSRFortessa™ X-20) to gate for apoptotic and live cells (see Supplementary Fig. 11). Cells transduced with shRNA sequences were assessed 72 h after transduction.

## Adhesion assay

96 well plates were coated with fibronectin (40ug/ml in PBS) for 1 h at 37 °C and blocked with blocking buffer (0.5% BSA in RPMI 1640) for 1 h at 37 °C. $2 \times 10^4$ KMS-11 cells were plated per well and incubated for 6 h at 37 °C. Non-adherent cells were removed by washing twice with PBS. Cells were stained with Crystal Violet dye (0.2% crystal violet in methanol) for 15 min at RT, washed in distilled water and dried overnight. Stained cells were lysed in 100 μl 2% SDS and absorbance at 590 nm was measured using an Infinite M200Pro plate reader (Tecan).

## Reporting summary

Further information on research design is available in the Nature Portfolio Reporting Summary linked to this article.

## Data availability

The publicly available epigenomic data used in this study are available in the NCBI database under accession code PRJEB25605[19], PRJNA608681[20] as well as [ftp://ftp.ebi.ac.uk/pub/databases/blueprint/releases/20160816/homo_sapiens] and [http://resources.idibaps.org/paper/chromatin-activation-as-a-unifying-principle-underlying-pathogenic-mechanisms-in-multiple-myeloma][21,22]. MMRF CoMMpass data can be accessed from the MMRF Researcher Gateway [https://research.themmrf.org/]. Cell line p52 KD RNA-seq (KMS-11 and MM1.144), p52 KD ATAC-seq (KMS-11), p52 KD H3K27ac HiChIP (KMS-11), p52 KD H3K27ac ChIP-seq (KMS-11, MM1.144 and LP1), endogenous H3K27ac and p52 ChIP-seq (KMS-11, JJN3, LP1, MM1.S, U266, MM1.144) raw and processed data generated in this study as well as the RNA-seq derived from NUHS patients have been deposited in the GEO database under accession code GSE230526. The remaining data are available within the Article, Supplementary Information or Source Data file. Source data are provided with this paper.

## Code availability

All computational analyses were performed using publicly available programmes as described in the Methods and Reporting Summary.

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

## Acknowledgements

This study is funded by the National Research Foundation (NRF) Singapore, under its Singapore NRF Fellowship (NRF-NRFF2018-04). These data were generated as part of the Multiple Myeloma Research Foundation Personalized Medicine Initiatives (https://research.themmrf.org and www.themmrf.org). In addition, we thank the Nanyang Assistant Professorship (NAP) Start-up-grant to Y.L.'s lab, Nanyang Technological University for the PhD scholarship funding of D.A.A. and National Medical Research Council (NMRC-OFYIRG16nov044) for their support. We appreciate assistance from the members of Y.L.'s lab who participated in this work and computing facilities from the HPCC (NTU) and National Supercomputing Centre Singapore. We thank Ms Myra Goh Shi Ying from IMCB, A*STAR for technical assistance in the orthotopic tumour xenograft studies.

## Author contributions

Y.L. conceptualized and supervised the study. Y.L. and D.A.A. planned and devised the experiments. J.M.C. designed and performed all computational analyses and data visualization, with input from N.H. D.A.A. and K.D. performed all molecular and cell biology experiments. D.A.A., J.H.T. and Q.C. contributed to the tumour xenograft studies. J.Z. and W.J.C. contributed to the RNA-seq data from M.M. patients. Y.L. and J.M.C. co-wrote the manuscript with input from D.A.A., K.D. and N.H.

## Competing interests

The authors declare no competing interests.
