## [Peer Review File · Nature Communications]

Aberrant non-canonical NF- κ B signalling reprograms the epigenome landscape to drive oncogenic transcriptomes in multiple myelomaReviewers' Comments:

Reviewer #1:

Remarks to the Author:

The current study entitled "Aberrant non-canonical NF- κ B signaling reprograms the epigenome landscape to drive oncogenic transcriptomes in multiple myeloma" by Ang DA et al. is aimed at determining the role of non-canonical NF- κ B and p52 in the recruitment of de novo enhancer activities, which are considered to be critical for the sustenance of multiple myeloma (MM) cell growth. While authors did a great job in compiling the genetic and epigenetic information from patient datasets and different MM cells using their advanced bioinformatic skill, there are several issues that has no been clearly addressed or not addressed at all. This somehow reduces the overall enthusiasm for the work in its current form. I have summarized some of my concerns are as follows:

1. From the capitulated findings in literature, authors described that MM subgroups having t(14;16), MAF, HRD with low TP53, and gain in chr.1q have the highest enrichment in NF- κ B+ samples. Accordingly, it was hypothesized that NF- κ B/p52 activation in MM cells is critical for recruitment of disease-specific enhancers. However, the authors undertook their NFKB2 KD and downstream experiments in KMS-11 cell line, which has primarily the t(4;14) translocation, but secondary t(14;16), and t(8;14) translocations. Therefore, the cell line is expected to harbor active enhancers from different cytogenetic aberrations. Therefore, it is definitely not the best representative cell line of the high-risk patient subgroups having over-expression of NF- κ B. It would have been best to perform these set of experiments in a cell line, which have single known t(14;16) such as ARD, NAN1 etc..
2. NF- κ B/p52 ChIP was performed in conjunction with the binding distribution of H3K27ac in different cell lines, such as MM.1S, U266 having high-risk, having intermediate-risk t(11;14), LP1, t(4;14) having standard, and KMS-11/JJN3 having multiple translocations and undetermined risk-staging. Nonetheless, authors found 17310 p52/K27ac binding sites, which were consistent across the lines irrespective of their underlying molecular make up. This raises a question that if p52 bound enhancers are consistent in both high-risk and intermediate-risk subgroups, how effectively this recapitulates the cytogenetic classification of the disease.
3. This is not clear, whether out of 17,310 p52 binding sites, only 657 p52 loci showed significant enhancer activities in the patient samples. If this is true, what is the explanation for this disparity is missing or ambiguous in the text.
4. It is not clear whether H3K4me1 marks were considered in conjunction with the H3K27ac for determining the poised enhancers.
5. It is important to validate the enhancer activities using some reporter assays, particularly for the top ones demonstrating p52 binding.
6. Considering the authors believe that NF- κ B/p52 takes part in 'chromatin remodeling; it would be important to define the topologically associating domains (TADs) in NFKB2 KD samples with the inclusion of CTCF binding distribution.
7. It would also be interesting to evaluate whether the dormant enhancers accrue any repressive marks such as H3K27me3 etc.
8. It is not clear what would be the practical implication of the findings in the current study. Authors need to explain, how these findings could be useful in clinics or in developing therapeutics.

It is highly recommended that authors carry out the suggested experiments and/ or prepare rebuttal to the critiques and resubmit. The manuscript may not be accepted in its current form.

Reviewer #2:

Remarks to the Author:

Ang et. al. profiled the relationship between multiple myeloma and NF- κ B pathway, by targeting disruption of p52 and showing that activation of p52 recruits cis-regulatory enhancers and super-enhancers resulting in changes in the 3D interactome. The manuscript is well written and the

experimental steps along with the conclusions are logical. However, I have some suggestions for this manuscript, specifically related to the looping data, which should be addressed before the manuscript can be formally accepted for publication.

Major Comments

1. In Figs. 4C-4D the authors highlighted BCL2 and IL6ST loci proximal to the p52 bound super enhancers. In Fig. 6 and Suppl. Fig. 6, they further deleted proximal SEs to show the change in BCL2 / RGS1 expression. Here, the utility of the HiChIP loops (or in general, EPIs) are not considered, where genes looping to particular enhancers with differential chromatin signal (between two conditions) may reveal significant changes in their expressions.

a. Can the authors quantify differential ChIP-seq peaks (specifically p52 KD peaks) and genes linked via EPI (HiChIP looping genes), preferably by differential HiChIP loops between two conditions?

b. Further, they need to quantify the differential (or p52 KD) super-enhancers and genes looping to these differential super-enhancers having differential expression between conditions and show example loci. I understand that Fig. 5e depicts some genes, but a detailed locus-specific example with HiChIP loops, genes, and SE is required.

2. Does the P52 knockdown alter TAD organization? Authors can use the tool HPTAD (<https://doi.org/10.1016/j.csbj.2023.01.003>) to call TADs from HiChIP data, and identify differential TADs (using the tools like TADCompare: <https://doi.org/10.3389/fgene.2020.00158> or hicDifferentialTAD: <https://hicexplorer.readthedocs.io/en/latest/content/tools/hicDifferentialTAD.html>) between conditions, and check their overlaps with the differential SEs, to identify the reorganized chromatin region.

3. The example genes BCL2, and RGS1 have proximal SEs whose editing changes gene expression (Suppl. Fig. 6). However, there can be genes with looped distal SEs (without proximal SEs) and genes having both proximal and looped distal SEs. Can the authors identify genes having both a proximal SE P (like the example genes BCL2, and RGS1) and simultaneously a distal SE D such that there exists a differential HiChIP loop between that gene to D? Specifically, can the authors highlight genes with the following conditions?

a. P is differential but D is non-differential between conditions.

b. P is non-differential but D is differential.

c. Both P and D are differential SEs between conditions

In each case, what are the changes in gene expression? Does the proximal or distal looped SE have a higher effect on the gene expression?

Minor Comments

1. Fig. 6(a) legend – should be RGS1 locus instead of BCL2 locus.

2. The genome browser tracks need to highlight the differential regulatory regions, differential SEs, and differential loops. The quality of figures (for these tracks) needs to be considerably improved.

Reviewer #3:

Remarks to the Author:

Major comments

In Fig 1a, the authors analyzed NF-kB (-) vs NF-kB (+) group, including both canonical and non-canonical pathways. The authors then further analyzed only the non-canonical pathway (p52). What is the rationale to focus only on non-canonical pathway?

In Fig 1, The authors employed only one MM cell line (KMS-11) to analyze transcriptome mediated by the non-canonical pathway. The authors should use additional MM cell lines to strengthen the results.

In Fig2, the authors use p52 binding and its association with only H3K27-ac to identify enhancer

activation. Please employ other enhancer activation markers including H3K4-me3 and pol-II.

The authors demonstrated RGS-1 SE deletion induces downregulation of RGS-1 expression and apoptotic cell death in MM cells. It is very important to confirm the RGS-1-specific effect using RGS-1 KD/KO approach.

The authors should analyze the correlation of RGS1 and p52 in NF-kB+ vs NF-kB- patient samples to confirm the significance of RGS1 regulated by non-canonical NF-kB pathway in myeloma progression.

It seems there is no significant difference between each group in Fig 6h and 6j, as well as Supplementary Fig 6m. Please put the p-value in those figures.

In Fig6, the authors described that "RGS-1 is a novel target of p52 dependent cis-acting super-enhancer that drives myeloma progression via JNK and p38a signaling". The authors should demonstrate more data of contribution of JNK and p38a signaling pathways to support the claim.

Minor comments

Please explain what NF-kB (M) means in the figure legend.

The manuscript was not well written. There are several grammatical errors and typos, which should be carefully edited.

Revisions & Responses

Reviewer A Comments

Reviewer #1, expertise in MM genomics, epigenomics and super-enhancers (Remarks to the Author):

The current study entitled “Aberrant non-canonical NF-κB signaling reprograms the epigenome landscape to drive oncogenic transcriptomes in multiple myeloma” by Ang DA et al. is aimed at determining the role of non-canonical NF-κB and p52 in the recruitment of de novo enhancer activities, which are considered to be critical for the sustenance of multiple myeloma (MM) cell growth. While authors did a great job in compiling the genetic and epigenetic information from patient datasets and different MM cells using their advanced bioinformatic skill, there are several issues that has no been clearly addressed or not addressed at all. This somehow reduces the overall enthusiasm for the work in its current form. I have summarized some of my concerns are as follows:

Comment A1

1. From the capitulated findings in literature, authors described that MM subgroups having t(14;16), MAF, HRD with low TP53, and gain in chr.1q have the highest enrichment in NF-κB+ samples. Accordingly, it was hypothesized that NF-κB/p52 activation in MM cells is critical for recruitment of disease-specific enhancers. However, the authors undertook their NFKB2 KD and downstream experiments in KMS-11 cell line, which has primarily the t(4;14) translocation, but secondary t(14;16), and t(8;14) translocations. Therefore, the cell line is expected to harbor active enhancers from different cytogenetic aberrations. Therefore, it is definitely not the best representative cell line of the high-risk patient subgroups having over-expression of NF-κB. It would have been best to perform these set of experiments in a cell line, which have single known t(14;16) such as ARD, NAN1 etc..

Response A1

It is indeed important to perform investigations in cell line models representative of the disease under study to facilitate translational impact and expediate any clinical benefits. As we have presented, constitutive activation of the non-canonical NF-κB pathway arises through secondary mutations affecting up to 40% of patients. After analysing one of the largest myeloma datasets available (CoMMpass) we noted that patients exhibiting a high NF-κB index are strongly associated with 3 transcriptional subtypes defined in Skerget et al., 2021. High NF-κB patients were most significantly overrepresented in the **1q gain** group followed by **MAF** and **HRD, low TP53**. According to Skerget et al. all three patient groups reached median OS and are characterised by the following associations:

- **1q gain**
 - Enriched in 1q gain (74%)
 - Enriched in *TRAF3* LOF (loss of function)
 - Diminished in *NRAS* GOF (gain of function)
 - No clear association with any primary translocations
 - Mixture of hyperdiploidy and non-hyperdiploidy
- **MAF**
 - Enriched in 1q gain (61%)
 - Enriched in MAF immunoglobulin translocations
 - t(14;16)-MAF

- t(8;14)-MAFA
 - t(14;20)-MAFB and t(20;22)-MAFB
- Enriched in *MAF* GOF
- Enriched in *BIRC2*, *ATM*, *PARP4*, *CDK8*, *WWOX* and *SAMHD1* LOF
- **HRD, low TP53**
 - Enriched in hyperdiploidy
 - Enriched activity of NF-κB genes (*BCL10*, *TNFAIP3*, *IL8*, *GADD45B*, *NFKNIE*, *TNFIP1*, *NFKBIZ*, *IL2RG*, *CD40*, and *CD74*)
 - Diminished activity of *TP53*¹

All 3 patient subgroups exhibit divergent cytogenetic background associations or absence thereof. However, they all exhibit secondary associations linked to increased NF-κB activity (underlined) and the 2 groups with the strongest NF-κB+ enrichment (1q gain and MAF) were associated with secondary 1q gain events. Accordingly, the selection criteria for representative multiple myeloma cell lines (MMCLs) were defined as requiring a ncNF-κB activating mutation, particularly the TRAF3 mutation which is the most commonly identified mutation in patients and 1q gain secondary events. KMS-11 MMCL was selected as it effectively meets the selection criteria but also harbours the P53 loss of function mutation and produces consistent results upon p52 knock down.

Table 1. Known genetic and cytogenetic characteristics in multiple myeloma cell lines

MMCL	Primary events		Secondary events					Subtype	Risk
	Translocation	Gene Ex.	1q	P53	TRAF3	NIK	NRAS		
ANBL-6	t(14;16) ²	+MAF	WT*	LOF ²	LOF ²	WT*	WT*	MAF	High
JJN-3	t(14;16) ² t(8;14) ²	+MAF +MAFA	Gain ³	LOF ²	WT	GOF ⁴	WT*	MAF/1q	High
KMS-11	t(4;14) ² t(8;14) ² t(14;16) ²	+NSD2 +MAFA +MAF	Gain ³	LOF ²	LOF ²	WT*	WT ²	MS/MAF/ 1q	High
L363	t(20;22) ² t(11;14) ⁵	+MAFB +CCND1	Gain ³	MU ²	WT ⁶	GOF ⁴	LOF ²	MAF/1q	High
LP-1	t(4;14) ² t(8;14) ⁷	+NSD2 +FGFR3 +MAFA	Gain ³	LOF ²	LOF ²	WT*	WT ²	MS/MAF/ 1q	High
MM1.144	t(14;16) ² t(8;14) ²	+MAF +MAFA	Gain [†]	WT*	LOF ²	WT*	WT ²	MAF/1q	High
MM1.S	t(14;16) ² t(8;14) ²	+MAF +MAFA	Gain ³	WT ²	LOF ²	WT*	WT ²	MAF/1q	High
U-266	t(11;14) ²	+CCND1	Gain ³	LOF ²	LOF ²	WT*	WT ²	CD1/1q	High

† personal communication; * unconfirmed

Very few of the MMCLs available to us have a single t(14;16)-MAF translocation as requested, most harbour multiple translocations and MAF activation (see Table 1). The cell lines suggested, ARD and NAN1 are not known to have any ncNF-κB activating mutations and therefore do not meet the selection criteria. The only MMCL with a single t(14;16)-MAF translocation, ANBL-6, has unknown 1q status and must be cultured with the addition of IL-6 which is likely to activate the canonical NF-κB pathway – confounding our analyses of the non-canonical pathway. KMS-11 does indeed harbour a t(4;14)-NSD2 translocation alongside t(14;16)-MAF and t(8;14)-MAFA translocations, however these still result in MAF activation and several publications have reported MAF specific findings are reproducible in KMS-11^{8,9}. Additionally, Skerget et al., reported that a patient displaying both t(14;16)-MAF and t(4;14)-NSD2 translocations was still classified in the MAF subtype, suggesting the MAF transcriptional profile overrides the t(4;14) associated MS transcriptional profile¹⁰. KMS-11,

JJN3, MM1.S and MM1.144 are thus still the next best representative MMCLs for that particular MAF translocation compatible with the experimental design.

Variability of enhancer landscapes and other characteristics between cell lines is always a concern irrespective of cytogenetics, which is why we had already taken steps to validate the p52 regulated *RGS1* super enhancer and its impact on *RGS1* expression in additional cell lines: LP1 and JJN3. LP1 also harbours a mixture of t(8;14)-MAFA and t(4;14)-NSD2 translocations while JJN3 harbours only MAF specific t(14;16)-MAF and t(8;14)-MAFA translocations and the findings support what was originally identified in KMS-11. Additionally, we have revised our p52 and enhancer profiling experiments to include another cell line: MM1.144 which features t(14;16)-MAF and t(8;14)-MAFA translocations, further corroborating our initial findings in KMS-11 (Lines: 142-155, 174-193, 250-252, 502, 663, 686-713, 715-723; Figures: 2a-c, f-g; Supp. Figures: 1-4, 8 of the revised manuscript). Finally, through additional comparative analyses of the revised enhancer landscapes for MAF translocation specific MMCLs (t(14;16) and t(8;14)): JJN3, MM1.S and MM1.144, we identified 9328 common enhancers of which 94% are recapitulated in KMS-11 (Figure 1 herein). This indicates the core enhancer landscape of the MAF subtype is very well represented in KMS-11.

Figure 1. Enhancer landscapes across multiple myeloma cell lines carrying t(14;16)
Distal H3K27ac peaks detected across 4 MMCLs carrying the t(14;16) translocation. Common overlap highlighted in yellow. Interestingly, although MM1.S and MM1.144 have identical cytogenetics their enhancer landscapes still vary considerably.

Comment A2

2. NF- κ B/p52 ChIP was performed in conjunction with the binding distribution of H3K27ac in different cell lines, such as MM.1S, U266 having high-risk, having intermediate-risk t(11;14), LP1, t(4;14) having standard, and KMS-11/JJN3 having multiple translocations and undetermined risk-staging. Nonetheless, authors found 17310 p52/K27ac binding sites, which were consistent across the lines irrespective of their underlying molecular make up.

This raises a question that if p52 bound enhancers are consistent in both high-risk and intermediate-risk subgroups, how effectively this recapitulates the cytogenetic classification of the disease.

Response A2

We reported 17310 unique p52 binding sites consistent **within** each MMCL, not common across all MMCLs. There are only 1495 p52 binding sites found to be consistent across all MMCLs (supported by 2 replicates in each MMCL) in the initial analysis. In the revised analyses including MM1.144 there are 1389 p52 binding sites found to be consistent across all MMCLs (supported by 2 replicates in each MMCL). Additionally, further restriction of p52 binding sites to intergenic/intronic H3K27ac peaks (supported by 2 replicates in each MMCL) limits the total unique p52/K27ac sites to 6367 of which 380 are common across all MMCLs (Figure 2a herein). We have revised the second results section in the manuscript to communicate these numbers more clearly (Lines: 174-198; Figures: 2; Supp. Figures: 2 of the revised manuscript).

Figure 2. Comparison of p52 binding sites occurring in non-promoter H3K27ac peaks

a) Summary of reproducible p52 binding detected within intergenic or intronic H3K27ac peak regions across 6 multiple myeloma cell lines. 6367 total unique sites are detected with 380 common across all MMCLs. b) Hierarchical clustering of reproducible p52-bound H3K27ac peaks in low, medium and high risk MMCLs.

It is unclear what risk stratification strategy is being proposed to classify the MMCLs in question as they all harbour mutations or cytogenetics (p53 LOF, 1q gain, t(4;14), t(14;16) or t(14;20)) considered “high risk” according to several other sources^{11,12}. Nevertheless, under the assumption MM1.S/MM1.144 are high risk, U266 is intermediate risk and LP1 is standard as suggested – a basic comparative analysis of p52-bound enhancers seems to indicate the LP1 profile differs substantially from the higher risk MMCLs (Figure 2a and b). However, our study was not aimed or designed to examine the relationship between p52 enhancer activation and cytogenetic aberrations. Additional work beyond the scope of this study would be required to thoroughly examine if constitutively bound p52 enhancer landscapes are influenced by the complex cytogenetic aberrations that occur in the early stages of myelomagenesis and are routinely used for risk classification of patients. Our study was aimed at identifying what novel cis-regulatory networks are activated during constitutive activation of κ B in multiple myeloma resulting from secondary mutations that are enriched in multiple subtypes exhibiting poor survival.

Comment A3

3. This is not clear, whether out of 17,310 p52 binding sites, only 657 p52 loci showed significant enhancer activities in the patient samples. If this is true, what is the explanation for this disparity is missing or ambiguous in the text.

Response A3

Indeed, in the initial analysis 17310 unique p52 binding sites were identified. However, only 1495 were detected as commonly bound by p52 in all MMCLs and 7779 p52 binding sites unique to any 1 one of the 5 MMCLs. Thus, there is substantial variation between each MMCL and by extension each patient, reflecting the heterogenous nature of multiple myeloma. In the revised analysis, including MM1.144, there are 741 genome wide p52 loci showing consistent increased H3K27 acetylation across NF- κ B+ patients. Figure 2f-g have been adjusted to display the results relevant to sites falling in intergenic and intronic regions only (Lines: 174-198, 215-225; Figures: 2a-c, 2f-g; Supp. Figures: 2a-d, 2g of the revised manuscript). The most striking result we highlight through Figure 2f is the high proportion of dormant enhancers (de novo/re-activated) activated in myeloma which are annotated alongside the heatmap.

Comment A4

4. It is not clear whether H3K4me1 marks were considered in conjunction with the H3K27ac for determining the poised enhancers.

Response A4

We did not explicitly refer to any poised enhancers in the original manuscript. We only explained strong enhancers correspond to chromHMM states defined by the presence of H3K4me1 in conjunction with H3K27ac.

However, we have profiled H3K4me1 marks in KMS-11 to address Comment C3 in the revised manuscript. 97% of the H3K27ac peaks considered as enhancers (intergenic or intronic) in KMS-11 overlap H3K4me1 peaks. H3K4me1 signals are now also included in the genome tracks (Lines: 194-198, 542, 999; Supp. Figures: 2e, 4, 7 and 8 of the revised manuscript).

Comment A5

5. It is important to validate the enhancer activities using some reporter assays, particularly for the top ones demonstrating p52 binding.

Response A5

We have performed luciferase reporter assays to validate the enhancer activities of p52-associated 1 kb regions within the super-enhancers of *BCL2*, *RGS1* and *IL6ST*. All three p52-bound enhancers showed significant increase in luciferase activity following p52 overexpression, confirming the role of p52 in regulating their activity (Figure 3 herein). These results have been included in the revised manuscript as Supplementary Figure 9a (Lines: 367-370, 765-782 of the revised manuscript).

Figure 3. Ectopic expression of p52 drives the super-enhancer activity of myeloma essential genes

Relative luciferase activity of the p52-bound constituent enhancer of *BCL2*, *RGS1* or *IL6ST* SE regions in 293T cells with or without p52 overexpression. Fold luminescence is calculated by normalising luciferase luminescence reading by renilla luminescence reading. The normalized luciferase activity value in p52 overexpressing cells (fold activation) is then calculated as a fold change to the normalized GFP luciferase activity. Error bars represent mean \pm SD of three experimental replicates. *P<0.05, **P<0.01, ***P<0.001, ****P<0.0001, ns: not significant

Comment A6

6. Considering the authors believe that NF-kB/p52 takes part in 'chromatin remodeling; it would be important to define the topologically associating domains (TADs) in NFKB2 KD samples with the inclusion of CTCF binding distribution.

Response A6

In order to define the topologically associating domains (TADs) in NFKB2 KD samples, we followed the suggestions from Reviewer B (Comment B3). Briefly, we were able to call 4217 TADs from our HiChIP data at a 50KB resolution using HiCExplorer. Additionally, we performed CTCF ChIP-seq on control and NFKB2 KD samples. We observed general enrichment of CTCF peaks at the TAD boundaries but we did not detect any significant changes in CTCF binding upon NFKB2 KD (Figure 4 herein). However, up to 21% of the TADs detected showed significant inter or intra-TAD changes, suggesting that minor topological changes may be occurring independently of CTCF binding (Lines: 300-302, 316-320, 734-738; Supp. Figures 6c and 6f of the revised manuscript).

Our initial chromatin remodelling statements were based solely on the observation that histone acetylation, accessibility and interactions vary greatly in response to NFKB2 KD and did not encompass topological chromatin organisation. The machinery supporting enhancer-promoter regulation remains incompletely understood. However, it seems clear that CTCF-independent machinery exists¹³. Additionally, recent studies indicate CTCF or indeed

chromatin topology itself may not always be essential for maintaining cis-regulatory relationships and may be bypassed by other mechanisms involving other proteins, like cohesin^{14,15}.

Figure 4. CTCF binding at topologically associating domains and boundaries under NFKB2 KD
Profile and heatmap visualisations of CTCF binding at TADs and their boundaries. Signal normalised to 1X genome coverage. Control and NFKB2 knock down conditions are indicated in blue and green respectively in the profile plots (top) and in the heatmaps below with signal intensity indicated by red-blue gradient.

Comment A7

7. It would also be interesting to evaluate whether the dormant enhancers accrue any repressive marks such as H3K27me3 etc.

Response A7

Dormant enhancers as defined in the manuscript – correspond to *de novo* and re-activated enhancers. These are enhancers which have shown MM specific activity and are normally dormant in plasma cells. Previous B-cell activity distinguishes re-activated from *de novo* (new and specific to MM).

H3K27me3 and H3K27ac are mutually exclusive marks indicative of repressive or active regulatory activity respectively. Endogenous H3K27me3 ChIP-seq data for KMS-11 indeed confirms 99% of the H3K27ac peaks designated as enhancers in our analyses do not overlap H3K27me3 peaks (Figure 11 herein) (Lines: 194-198, 542, 553-556; Supp. Figure 2e of the revised manuscript).

Dormant enhancers may be expected to accrue repressive marks in absence of p52 stimulation, however it would be highly unlikely such repressive marks would be deposited in

an experimental setting after knock down of p52 in KMS-11 within a viable time frame. However, we did hypothesise deposition of H3K27me3 and other marks may be detected at these loci in non-myeloma tissue types. *De novo* enhancer activity across multiple tissue types found in the Epigenomics Roadmap collection are depicted in Figure 5f (original/revised) of the manuscript. The 18-state chromHMM model relies on H3K27me3 peaks to assign states 16 and 17 which are coloured in dark blue; 78% of the *de novo* loci show H3K27me3 deposition in at least one of the tissue types.

Comment A8

8. It is not clear what would be the practical implication of the findings in the current study. Authors need to explain, how these findings could be useful in clinics or in developing therapeutics.

Response A8

Our findings highlight constitutive $\text{NF-}\kappa\text{B}$ activity initiated by activating mutations has a significant impact on myeloma progression via its modulation of the myeloma enhancer landscape. Therefore, patient risk classification may benefit from additional screening for TRAF3 mutations or increased $\text{NF-}\kappa\text{B}$ activity especially if they have tested positive for MAF and 1q gain events. More specific targeting of the non-canonical pathway may also provide new diagnostic and treatment strategies for patients. Our study has already characterised the roles of several $\text{NF-}\kappa\text{B}$ target genes including RGS1 and BCL2. In addition to the reports on activated p38/MAPK14 negatively regulating the activity of JNK signalling pathway, our data suggests RGS1 is an important factor regulating the active state of p38/MAPK14 which in turn regulates GRAP2 levels and activity towards a pro-proliferative signalling cascade in MM. These target genes can potentially act as biomarkers towards screening and targeted therapeutic approaches in $\text{NF-}\kappa\text{B+}/\text{MAF+}/1\text{q}$ gain patients. Especially considering pharmacological inhibitors for other GTPase family members have been identified¹⁶. We have updated the discussion in the revised manuscript with these points in mind (Lines: 473-490 of the revised manuscript).

Comment A9

It is highly recommended that authors carry out the suggested experiments and/ or prepare rebuttal to the critiques and resubmit. The manuscript may not be accepted in its current form.

Response A9

Thank you for your feedback, we hope our response and revised manuscript addresses your concerns sufficiently.

Reviewer B Comments

Reviewer #2, expertise in epigenetics (Remarks to the Author):

Ang et. al. profiled the relationship between multiple myeloma and $\text{NF-}\kappa\text{B}$ pathway, by targeting disruption of p52 and showing that activation of p52 recruits cis-regulatory enhancers and super-enhancers resulting in changes in the 3D interactome. The manuscript is well written and the experimental steps along with the conclusions are logical. However, I have some suggestions for this manuscript, specifically related to the looping data, which should be addressed before the manuscript can be formally accepted for publication.

Major Comments

Comment B1

1. In Figs. 4C-4D the authors highlighted *BCL2* and *IL6ST* loci proximal to the p52 bound super enhancers. In Fig. 6 and Suppl. Fig. 6, they further deleted proximal SEs to show the change in *BCL2* / *RGS1* expression. Here, the utility of the HiChIP loops (or in general, EPIs) are not considered, where genes looping to particular enhancers with differential chromatin signal (between two conditions) may reveal significant changes in their expressions.

Response B1

Although the HiChIP data is not illustrated in Figure 4c and d of the original manuscript, we would like to clarify that the HiChIP data has been fully integrated into our analyses with enhancers and genes as illustrated in Figure 5 onwards. Figures 5b, c, d and e in particular highlighted the Enhancer-Promoter pairings directly supported by HiChIP data. Furthermore, key differential loops were illustrated in Figure 6a and Supp. Figure 6a genome track browser views. *BCL2* and *RGS1* SEs were not selected based on their proximity to the genes but based on the significant concordant changes in enhancer, promoter and interaction dynamics. Both cases have multiple enhancers/genes in their vicinity. We have now revised these visualisations to feature improved readability of the SE and HiChIP tracks as requested in the minor comments. The HiChIP data is also incorporated in the genome browser tracks at 2 levels: Firstly, the contact matrix features differential loops represented as squares coloured by log₂ fold change between p52 KD and control conditions. Secondly, differential loops are represented as arcs coloured according to counts per million (Figures: 4c-d, 6a; Supp. Figures: 4, 7a-c and 8 of the revised manuscript).

Comment B1.1

a. Can the authors quantify differential ChIP-seq peaks (specifically p52 KD peaks) and genes linked via EPI (HiChIP looping genes), preferably by differential HiChIP loops between two conditions?

Response B1.1

As requested, we have quantified more clearly the differential H3K27ac ChIP-seq peaks and genes linked by EPI. These are presented as a table and a simplified alluvial plot (Table 2 and Figure 5 herein) (Supp. Table 2; Supp. Figure 6e of the revised manuscript). For this query, we can report that 357 unique downregulated enhancers interact with 351 unique downregulated gene promoters via 543 downregulated loops, this results in a total of 695 unique Enhancer-Promoter-Interaction combinations undergoing significant downregulation across all 3 features after p52 knock down.

Comment B1.2

b. Further, they need to quantify the differential (or p52 KD) super-enhancers and genes looping to these differential super-enhancers having differential expression between conditions and show example loci. I understand that Fig. 5e depicts some genes, but a detailed locus-specific example with HiChIP loops, genes, and SE is required.

Response B1.2

As requested, we have quantified more clearly the differential H3K27ac ChIP-seq peaks and genes linked by EPIs. These are presented as a table and a simplified alluvial plot (Table 2 and Figure 5 herein). For this query focusing on super enhancers (SEs), we can report 499 unique downregulated SE constituents were called. 127 of these unique downregulated SE constituents interact with 116 unique downregulated genes via 217 unique loops, this results in a total of 283 unique SE-Promoter-Interaction combinations undergoing significant downregulation across all 3 features after p52 knock down. As stated earlier in Response B1, locus specific examples depicting key differential HiChIP loops, genes and SEs are provided in Figure 6a, Supp. Figures: 7a-c and 8 of the revised manuscript . We have also included new loci for *IRF2BP2*, *CREG1* and *KIF21B*, featuring both proximal and distal SEs included in Supp. Figure 7 and herein (Figure 10).

Figure 5. Overview of enhancer-promoter dynamics in KMS-11 p52 KD

Alluvial plot highlighting differential features (Enhancers or Promoters) connected through significant differential interactions forming EPI pairs resulting from p52 knockdown in KMS-11 cells. Most downregulated features show downregulation in interactions with their downregulated counterparts.

Table 2. Summary of features across concordantly regulated EPI pairs

Features	Enhancers (Super-Enhancers)	Interactions	Genes	EPI Pairs
Upregulated	75 (19)	90 (24)	77 (20)	99 (26)
Downregulated	357 (127)	543 (217)	351 (116)	695 (283)

Comment B2

2. Does the P52 knockdown alter TAD organization? Authors can use the tool HPTAD (<https://doi.org/10.1016/j.csbj.2023.01.003>) to call TADs from HiChIP data, and identify differential TADs (using the tools like TADCompare: <https://doi.org/10.3389/fgene.2020.00158> or hicDifferentialTAD: <https://hicexplorer.readthedocs.io/en/latest/content/tools/hicDifferentialTAD.html>) between

conditions, and check their overlaps with the differential SEs, to identify the reorganized chromatin region.

Response B2

This is an interesting question that our manuscript did not directly address before. We thank Reviewer B for the helpful bioinformatics suggestions to assist with the enquiry. As recommended, we attempted to call TADs from HiChIP data using HPTAD however we were unable to get this specific program working with our data. We therefore opted to use the HiCEXplorer suite to both call TADs (hicFindTADs) and evaluate their dynamics (hicDifferentialTAD). We tested hicFindTADs at multiple resolutions including 2.5KB, 10KB, 25KB, 40KB and 50KB. We found that 50KB yielded the most convincing TAD calls, based on visual evaluation of several interaction rich regions (Figure 6 herein).

Figure 6. Examples of TAD calling at 10K and 50K resolutions

Contact matrices annotated with TADs detected by hicFindTADs at 10KB or 50KB resolutions. Upper matrices and lower matrices are extracted from the control and p52 knock down conditions respectively. TADs are coloured black if they are stable and grey if they are dynamic in the control matrices. Dynamic TADs are coloured in lower matrices highlighting Left-inter-TADs (Red), Intra-TADs (Green) and Right-inter-TADs (Blue). Combinations of Intra-TADs with Left-inter-TADs or Right-inter-TADs are coloured Yellow or Cyan respectively.

Utilising this approach 4217 domains were called from our HiChIP data. We found the TAD boundaries to be enriched in CTCF binding after investigation of CTCF ChIP-seq binding upon p52 KD as recommended in Comment A6 (Figure 4 herein). hicDifferentialTAD called 867 differential TADs (21%) of which 723 featured inter-TAD dynamics and 247 featured intra-TAD dynamics (Figure 7 herein). Most differential loops and SEs overlapped TADs without any detectable change in TAD dynamics (Figure 8 herein).

Figure 7. Summary of differential TADs detected

867 differential TADs were detected by hicDifferentialTAD representing 21% of all TADs called. Differential TADs are classified as undergoing inter (723) or intra (247) TAD changes or both (103).

Figure 8. Summary of TAD overlaps with differential loops and SEs

Numbers of TADs overlapping unchanged or differential loops and super enhancers (SEs). Differential loops or SEs do not appear to be enriched in differential TADs with the vast majority residing in unchanged TADs.

To understand how TAD dynamics might relate to other features such as SEs, we examined the overlaps between our EPI pair collection and the TADs. Most differential EP pairs occurred in stable TADs. However, 181 differential EP pairs occurred in differential TADs and 76 involved super enhancers (Figure 9 herein). Several of the SE linked genes reported in our study are implicated, notably RGS1. We can conclude then that p52 has some limited consequences on chromatin topological remodelling and that the RGS1 locus resides in a region undergoing such significant changes during p52 knockdown. Please refer to A6 for further discussion. Revisions applied to Lines: 300-302, 318-320, 734-736; Supp. Figures 6c and 6f of the revised manuscript.

Figure 9. Summary of Enhancer to Promoter interactions in the context of TAD changes

Alluvia are coloured to highlight dynamic interactions connecting downregulated enhancer and gene expression features (green). As shown by the alluvia, most dynamic EPI pairs occur in unchanged TADs however a minority do occur in TADs undergoing significant intra-TAD or/and inter-TAD changes (yellow).

Comment B3

3. The example genes BCL2, and RGS1 have proximal SEs whose editing changes gene expression (Suppl. Fig. 6). However, there can be genes with looped distal SEs (without proximal SEs) and genes having both proximal and looped distal SEs. Can the authors identify genes having both a proximal SE P (like the example genes BCL2, and RGS1) and simultaneously a distal SE D such that there exists a differential HiChIP loop between that gene to D? Specifically, can the authors highlight genes with the following conditions?

- P is differential but D is non-differential between conditions.
- P is non-differential but D is differential.
- Both P and D are differential SEs between conditions

In each case, what are the changes in gene expression? Does the proximal or distal looped SE have a higher effect on the gene expression?

Response B3

This is another interesting question we did not approach in the manuscript. As suggested, we defined enhancers residing within 10KB of a promoter as Proximal (P) and assigned enhancers linked to promoters by a HiChIP loop as Distal (D) (since these are only detected beyond 10KB distance). We then selected genes with a differential P, D or both producing P, D or PD groups. We found 1820 genes implicating enhancers meeting these criteria and 190 implicating super enhancers. There appears to be a certain synergy when both P and D enhancers are differential as the gene expression distribution for this category (PD) appears lower than genes with a single active distal or proximal enhancer (D or P) (Figure 10 herein). However more cases would be needed to obtain robust statistics for super-enhancers. Revisions applied to Lines: 335-341, Supp. Table 3 and Supp. Figure 7 of the revised manuscript.

Figure 10. Regulation by distal and proximal enhancers and super-enhancers

(a-c) Genome browser visualisations of 3 loci highlighting genes regulated through (a) Distal+Proximal, (b) Proximal or (c) Distal SEs. Each panel has 10 tracks; first 5 show ChIP/HiChIP-seq signal and peaks for p52, H3K4me1 and H3K27ac signal under normal/control conditions, next 3 show H3K27ac ChIP/HiChIP-seq signal and peaks under p52 knock down. Before last track shows STABULO classification for each significant peak. SEs are annotated with rectangles. Final track shows gene annotations. (d-e) Gene expression changes after p52 knockdown for genes linked to distal and proximal (d) enhancers or (e) super-enhancers undergoing significant loss of activity for at least one

enhancer. Genes with only distal enhancers undergoing loss are labelled Distal, genes with only proximal enhancers undergoing loss are labelled Proximal, genes with both distal and proximal enhancers undergoing loss are labelled Distal+Proximal.

Table 3. List of genes linked to proximal and distal super-enhancers

Group	Gene	log ₂ (FC)	FDR
Proximal	AL358473.1	-2.09	1.41E-02
Proximal	KIF21B	-1.61	6.03E-03
Distal	AC126696.3	-1.55	7.00E-02
Proximal	LAIR1	-1.28	4.90E-06
Distal	STOM	-1.08	6.58E-13
Proximal	LINC01686	-1.02	1.01E-02
Distal+Proximal	WIP1	-0.97	1.78E-02
Distal	LINC02362	-0.86	5.53E-07
Proximal	ERN1	-0.85	5.18E-06
Proximal	ANKRD36BP2	-0.80	1.80E-08
Distal	RHOD	-0.79	2.53E-04
Distal+Proximal	TMSB4X	-0.78	9.41E-12
Distal+Proximal	RGS16	-0.77	2.58E-07
Distal	IRF2BP2	-0.67	2.74E-07
Distal+Proximal	ADTRP	-0.67	2.42E-04
Distal+Proximal	AL022724.3	-0.66	1.90E-03
Proximal	AL360182.2	-0.65	7.58E-02
Distal	AL160408.2	-0.64	3.53E-05
Proximal	UBC	-0.63	4.46E-05
Distal	AL365272.1	-0.57	3.54E-04
Proximal	CD48	-0.51	6.00E-05
Distal+Proximal	CREG1	-0.51	2.57E-05
Proximal	MXI1	-0.46	1.53E-03
Distal+Proximal	WWC3	-0.45	1.68E-02
Distal	UBALD2	-0.42	1.52E-02
Distal+Proximal	DUSP22	-0.38	1.50E-02
Distal+Proximal	NDUFAF6	-0.38	3.94E-02
Distal+Proximal	QPCT	-0.38	5.38E-03
Proximal	SYNGR2	-0.35	4.26E-02
Proximal	NFIL3	-0.34	8.96E-02
Proximal	SUB1	-0.34	2.94E-02
Distal+Proximal	CYTIP	-0.32	3.21E-02
Proximal	GALM	-0.31	5.26E-02
Distal	PHF19	-0.30	7.20E-02
Distal	IL6ST	-0.28	4.97E-02
Distal+Proximal	SEPTIN6	-0.28	7.16E-02
Distal	FHL1	-0.26	8.89E-02

Minor Comments

Comment B4

1. Fig. 6(a) legend – should be RGS1 locus instead of BCL2 locus.

Response B4

Thank you, the typo has been corrected (line: 1003).

Comment B5

2. The genome browser tracks need to highlight the differential regulatory regions, differential SEs, and differential loops. The quality of figures (for these tracks) needs to be considerably improved.

Response B5

We have modified the genome browser tracks for improved legibility as requested (Figures: 4c-d, 6a; Supp. Figures: 4, 7a-c and 8 of the revised manuscript).

Thank you for your valuable feedback.

Reviewer C Comments

Reviewer #3, expertise in the non-canonical NF- κ B pathway in MM (Remarks to the Author):

Major comments

Comment C1

1. In Fig 1a, the authors analyzed NF- κ B (-) vs NF- κ B (+) group, including both canonical and non-canonical pathways. The authors then further analyzed only the non-canonical pathway (p52). What is the rationale to focus only on non-canonical pathway?

Response C1

Most studies have shown the effect of the classical NF- κ B pathway in regulating cancer and diseases. The causal involvement of ncNF- κ B signalling is understudied in the context of cancer epigenomics. Moreover, the link between ncNF- κ B activating mutations and chromatin activation in myelomagenesis has not been elucidated to date. Although a higher frequency of genetic alterations in the NF- κ B pathway has been reported in relapsed refractory multiple myeloma (RRMM) patients, nobody has profiled the role of ncNF- κ B pathway in driving MM progression through the epigenome¹⁷. TRAF3 mutations are highly enriched in the NF- κ B+ group and specifically affect the non-canonical NF- κ B pathway¹⁸.

Comment C2

2. In Fig 1, The authors employed only one MM cell line (KMS-11) to analyze transcriptome mediated by the non-canonical pathway. The authors should use additional MM cell lines to strengthen the results.

Response C2

As requested, we have performed RNA sequencing and transcriptomic analyses for an additional cell line undergoing p52 knock down: MM1.144. The MM1.144 transcriptome showed downregulation of a comparable number of genes deemed essential for multiple myeloma during p52 knock down. Additionally, the ATAC-seq and ChIP-seq experiments performed in MM1.144 clearly corroborate the original findings in KMS-11 (Lines: 142-155, 174-193, 250-252, 502, 663, 686-713, 715-723; Figures: 2a-c, f-g; Supp. Figures: 1-4, 8 of the revised manuscript).

Comment C3

- In Fig2, the authors use p52 binding and its association with only H3K27-ac to identify enhancer activation. Please employ other enhancer activation markers including H3K4-me3 and pol-II.

Response C3

As far as we know H3K4me3 and pol-II are primarily associated with promoter activity. Typically, H3K27ac and H3K4me1 are used to identify active enhancers^{19,20}. We have therefore opted to employ H3K4me1 to help address both this comment and Comment A3 to validate the H3K27ac marked loci which correspond to active enhancers. Over 97% of the p52-bound H3K27ac peaks located in intergenic or intronic regions overlap with H3K4me1 peaks in KMS-11. A summary of the ChIP-seq signals is now provided (Figure 11 herein) and H3K4me1 signals are now included for all genome browser visualisations in the revised manuscript (Lines: 194-198, 542, 552-556; Supp. Figures: 2e, 4, 7 and 8 of the revised manuscript).

Figure 11. ChIP-seq signals for p52-bound putative enhancers identified in KMS-11
ChIP-seq coverage signal profiles and heatmaps for p52, H3K27ac, H3K4me1 and H3K27me3 at p52-bound intergenic/intronic H3K27ac peaks obtained in KMS-11.

Comment C4

- The authors demonstrated RGS-1 SE deletion induces downregulation of RGS-1 expression and apoptotic cell death in MM cells. It is very important to confirm the RGS-1-specific effect using RGS-1 KD/KO approach.

Response C4

We previously performed shRNA mediated RGS-1 KD (data originally provided in Supp. Figures 6h and 6i, currently Supp. Figures 9g and 9h) to confirm the functional importance of RGS1 in mediating the enhanced proliferation of KMS-11 and LP1 cells. Additionally, as suggested, we have performed western blot analysis for the p38-JNK signalling cascade in RGS-1 KD cells presented in Figure 12 herein and included in the revised manuscript in Supp. Figure 9f.

Figure 12. Knock down (KD) of RGS1 alters p38-GRAP2-JNK signalling axis in MM cells

A: Representative western blot showing shRNA mediated KD of RGS1 enhances p-p38 level but decreases the expression and activation of GRAP2-pJNK signalling cascade, further downregulating the expression of its target proteins including cdc2 and cyclin D1 in MM cell lines (KMS11 and LP1). **B:** Graphs representing the densitometric quantification of the expression level of the targets studied in figure 11A. Normalization was done taking GAPDH as loading control and plotted relative to scramble (Taking scramble as 1). N=3 and error bar is SEM.

Comment C5

- The authors should analyze the correlation of RGS1 and p52 in NF- κ B⁺ vs NF- κ B⁻ patient samples to confirm the significance of RGS1 regulated by non-canonical NF- κ B pathway in myeloma progression.

Response C5

An analysis of the relationship between NF- κ B⁺ and NF- κ B⁻ was provided in Figure 6b of the original manuscript, higher levels of RGS1 are indeed associated with higher ncNF- κ B activity. There is also a significant positive linear correlation between patient RGS1 and p52 expression which we present in Figure 13 of this document.

Figure 13. Correlation of p52 and RGS1 expression in CoMMpass patients

Normalised log transformed counts for *NFKB2* and *RGS1* transcripts in the MMRF CoMMpass dataset indicate a significant positive correlation (Pearson).

Comment C6

6. It seems there is no significant difference between each group in Fig 6h and 6j, as well as Supplementary Fig 6m. Please put the p-value in those figures.

Response C6

As requested, we have detailed the p-values in Figure 6i-j and Supp. Figure 9n of the revised manuscript.

Comment C7

7. In Fig6, the authors described that “RGS-1 is a novel target of p52 dependent cis-acting super-enhancer that drives myeloma progression via JNK and p38a signaling”. The authors should demonstrate more data of contribution of JNK and p38a signaling pathways to support the claim.

Response C7

We have further examined the role of RGS1 in regulating the p38-Grp2-JNK signalling axis to exert proliferative functions in myeloma cells. Grp2 acts as one of the major adaptor proteins that activate the JNK pathway, leading to the induction of downstream pro-proliferative genes including cyclin D1 and *cdc2*²¹. In addition, p-p38/MAPK14 is known to inhibit the JNK pathway via targeting Grp2/HPK1^{22,23}. Hence, we investigated

whether RGS1 is involved in activation of the JNK pathway through inhibiting p-38/Mapk14 activation, which in turn augments Grap2 expression. In the revised manuscript, we have performed the following additional experiments and demonstrated the role of RGS1 in activating JNK signalling through p38 inhibition and Grap2 upregulation. We have also confirmed the activation of downstream JNK targets, cdc2 and cyclin D1, by RGS1 (Lines: 403-411, 473-478, 521-524; Figure 6g and Supp. Figure 9f of the revised manuscript).

Additional Experiments:

i. RGS1 over expression (OE) and shRNA knock-down (KD) of GRAP2 was performed in MMCLs, followed by immunoblot analysis of the expression of downstream and upstream pathway markers in the p38-Grap2-JNK signalling axis (Supp. Figure 9f of the revised manuscript).

A

B

Figure 14. Knockdown of GRAP2 adaptor protein attenuates the pro-proliferative role of RGS1.

A: Representative western blot showing shRNA mediated KD of GRAP2 attenuates the effect of RGS1 OE on the activation and expression level of JNK, further downregulating the expression of its target proteins including cdc2 and cyclinD1 in MM cell lines (KMS11 and LP1). The level of p-p38 remains the same, independent of GRAP2 level, suggesting p38 to be upstream of GRAP2. **B:** Graphs representing the densitometric quantification of the expression level of the targets studied in figure 13A. Normalization was done taking GAPDH as loading control and enrichment quantified value (AU) is plotted. N=3 and error bar is SEM.

ii. Addition of p38 inhibitor (Talmapimod) studies in MMCLs and correlation of the expression levels of p-p38 with GRAP2 expression.

Figure 15. Level of p-p38 is inversely correlated with the expression of GRAP2 adaptor protein.

A: Representative western blot showing treatment of MM cells with p38 inhibitor (Talmapimod, 1 μM for 1h) enhances the expression of GRAP2 protein. **B:** Graphs representing the densitometric quantification of the expression level of the targets studied in figure 14A. Normalization was done taking GAPDH as loading control and plotted relative to untreated talmapimod (-Talmapimod) samples. N=3 and error bar is SEM.

iii. Additional immunoblot analysis of GRAP2, cyclinD1 and p-cdc2/cdc2 amended to the original panel for Figure 6g (revised as Figure 6g in the revised manuscript).

Figure 16. Ectopic expression of RGS1 rescues the inhibition of JNK pathway upon RGS1 SE deletion through GRAP2 mediated induction of JNK phosphorylation.

Representative western blot showing activated expression level of GRAP2 and its downstream pro-proliferative target proteins including p-cdc2 and cyclinD1 in WT+pLJM-GFP, WT+pLJM-RGS1, RGS1 SE del+pLJM-GFP and RGS1 SE-del+pLJM-RGS1 MM cells (KMS11 and LP1). N=3

Here we also provide a summarised graphical abstract for p52 regulated SE in the activation of RGS1 expression, which in turn regulates MM progression via the p38/GRAP2/JNK axis (Figure 18 herein; Figure 6h of the revised manuscript).

Figure 17. Graphical abstract showing RGS1 assisted signalling pathway regulating MM progression via p38/GRAP2/JNK axis. Created with BioRender.com

Minor comments

Comment C8

8. Please explain what NF- κ B (M) means in the figure legend.

Response C8

These indicate samples with mutations affecting the NF- κ B pathway. We have revised the legend accordingly in Figure 1f.

Comment C9

9. The manuscript was not well written. There are several grammatical errors and typos, which should be carefully edited.

Response C9

We have carefully revised the manuscript to correct any typographical and grammatical errors.

Thank you for taking the time to review this work.

References

1. Jang, H. *et al.* The Tumor Suppressor, p53, Negatively Regulates Non-Canonical NF- κ B Signaling through miRNA-Induced Silencing of NF- κ B-Inducing Kinase. *Mol. Cells* **43**, 23–33 (2020).
2. Keats, J. J. HMCL Characteristics. *Keats Lab* <https://www.keatslab.org/myeloma-cell-lines/hmcl-characteristics>.
3. Menezes, K. *et al.* High-Throughput Molecular Cancer Cell Line Characterization Using Digital Multiplex Ligation-Dependent Probe Amplification for Improved Standardization of in Vitro Research. *J. Mol. Diagn.* **22**, 1179–1188 (2020).
4. Annunziata, C. M. *et al.* Frequent engagement of the classical and alternative NF- κ B pathways by diverse genetic abnormalities in multiple myeloma. *Cancer Cell* **12**, 115–130 (2007).
5. Chow, S. *et al.* Myeloma immunoglobulin rearrangement and translocation detection through targeted capture sequencing. *Life Sci Alliance* **6**, (2023).
6. Keats, J. J. *et al.* Promiscuous mutations activate the noncanonical NF- κ B pathway in multiple myeloma. *Cancer Cell* **12**, 131–144 (2007).
7. Bolli, N. *et al.* A DNA target-enrichment approach to detect mutations, copy number changes and immunoglobulin translocations in multiple myeloma. *Blood Cancer J.* **6**, e467 (2016).
8. Jia, Y. *et al.* Myeloma-specific superenhancers affect genes of biological and clinical relevance in myeloma. *Blood Cancer J.* **11**, 32 (2021).
9. Chesi, M. *et al.* Frequent dysregulation of the c-maf proto-oncogene at 16q23 by translocation to an Ig locus in multiple myeloma. *Blood* **91**, 4457–4463 (1998).
10. Skerget, S. *et al.* Genomic Basis of Multiple Myeloma Subtypes from the MMRF CoMMpass Study. *bioRxiv* (2021) doi:10.1101/2021.08.02.21261211.
11. Stewart, A. K. *et al.* A practical guide to defining high-risk myeloma for clinical trials, patient counseling and choice of therapy. *Leukemia* **21**, 529–534 (2007).

12. Rajkumar, S. V. Multiple myeloma: 2022 update on diagnosis, risk stratification, and management. *Am. J. Hematol.* **97**, 1086–1107 (2022).
13. Kojic, A. *et al.* Distinct roles of cohesin-SA1 and cohesin-SA2 in 3D chromosome organization. *Nat. Struct. Mol. Biol.* **25**, 496–504 (2018).
14. Chakraborty, S. *et al.* Enhancer-promoter interactions can bypass CTCF-mediated boundaries and contribute to phenotypic robustness. *Nat. Genet.* **55**, 280–290 (2023).
15. Hsieh, T.-H. S. *et al.* Enhancer-promoter interactions and transcription are largely maintained upon acute loss of CTCF, cohesin, WAPL or YY1. *Nat. Genet.* **54**, 1919–1932 (2022).
16. O'Brien, J. B., Wilkinson, J. C. & Roman, D. L. Regulator of G-protein signaling (RGS) proteins as drug targets: Progress and future potentials. *J. Biol. Chem.* **294**, 18571–18585 (2019).
17. Vo, J. N. *et al.* The genetic heterogeneity and drug resistance mechanisms of relapsed refractory multiple myeloma. *Nat. Commun.* **13**, 3750 (2022).
18. He, J. Q., Saha, S. K., Kang, J. R., Zarnegar, B. & Cheng, G. Specificity of TRAF3 in its negative regulation of the noncanonical NF-kappa B pathway. *J. Biol. Chem.* **282**, 3688–3694 (2007).
19. Alvarez-Benayas, J. *et al.* Chromatin-based, in cis and in trans regulatory rewiring underpins distinct oncogenic transcriptomes in multiple myeloma. *Nat. Commun.* **12**, 5450 (2021).
20. Ordoñez, R. *et al.* Chromatin activation as a unifying principle underlying pathogenic mechanisms in multiple myeloma. *Genome Res.* **30**, 1217–1227 (2020).
21. Ma, W. *et al.* Leukocyte-specific adaptor protein Grap2 interacts with hematopoietic progenitor kinase 1 (HPK1) to activate JNK signaling pathway in T lymphocytes. *Oncogene* **20**, 1703–1714 (2001).
22. Hui, L. *et al.* p38alpha suppresses normal and cancer cell proliferation by antagonizing the JNK-c-Jun pathway. *Nat. Genet.* **39**, 741–749 (2007).

23. Wagner, E. F. & Nebreda, A. R. Signal integration by JNK and p38 MAPK pathways in cancer development. *Nat. Rev. Cancer* **9**, 537–549 (2009).

Reviewers' Comments:

Reviewer #1:

Remarks to the Author:

The manuscript "Aberrant non-canonical NF- κ B signalling reprograms the epigenome landscape to drive oncogenic transcriptomes in multiple myeloma" by Ang et al. is a resubmission. The authors put significant effort into improving the quality and clarity of the manuscript. While the current version of the manuscript has covered many of the defects of the primary submission, I would suggest a couple of additional works or request clarifications to further improve the overall quality of the work. My comments are appended below:

Major comments

1. Authors evaluated how NF- κ B/p52-dependent enhancers impact the super-enhancer dynamics based on H3K27ac profiling in KMS-11 and primary tumors. Previously, Lovel et al. (Cell 2013) reported that co-occupancy of BRD4 and MED1 is critical in classifying enhancers from super-enhancers in myeloma. If authors are willing to correlate the NF- κ B/p52 and super-enhancer dependency, they are encouraged to include BRD4 and MED1 ChIP data and H3K27ac to annotate the super-enhancers in myeloma.

Minor comments

1. Page 3, line 100. Please change 'genomic expression' to mutational profile.
2. Page 5, lines 199-203: It is unclear what the authors meant by "enhancer activity during myelomagenesis". To my understanding, MM develops from the MGUS stage (Monoclonal gammopathy of undetermined significance) and transitions through SMM (smoldering multiple myeloma) to attain the MM stage. Since the authors are not capturing the developmental stages of the disease, I am unsure how justified or correct introducing a new classification system, STABILO, is.
3. Page 5, line 211: Are de novo and reactivated enhancers the same? What do the reactivated enhancers mean? Similarly, what does "co-opted from enhancers in B cells" mean? MM is a neoplasm of plasma B cells, not primary B cells. The heterogeneity is far more complicated than the way it is mentioned in the manuscript.
4. Page 6, line 221: Here, authors mention that "p52 loci were frequently associated with de novo and reactivated enhancers". This is highly confusing and confounding to the critique #3.
5. In figures, the panels that represent ChIP data, it would be nice to clarify the control in the legends.

Reviewer #2:

Remarks to the Author:

The authors have addressed my comments in detail, and revised the manuscript accordingly. I do not have any further comments and recommend accepting the manuscript.

Reviewer #3:

Remarks to the Author:

The authors successfully addressed to comments/suggestions and the manuscript is now acceptable.

Comment 0

Reviewer #1 (Remarks to the Author):

The manuscript “Aberrant non-canonical NF-κB signalling reprograms the epigenome landscape to drive oncogenic transcriptomes in multiple myeloma” by Ang et al. is a resubmission. The authors put significant effort into improving the quality and clarity of the manuscript. While the current version of the manuscript has covered many of the defects of the primary submission, I would suggest a couple of additional works or request clarifications to further improve the overall quality of the work. My comments are appended below:

Comment 1

Major comments

1. Authors evaluated how NF-κB/p52-dependent enhancers impact the super-enhancer dynamics based on H3K27ac profiling in KMS-11 and primary tumors. Previously, Loven et al. (Cell 2013) reported that co-occupancy of BRD4 and MED1 is critical in classifying enhancers from super-enhancers in myeloma. If authors are willing to correlate the NF-κB/p52 and super-enhancer dependency, they are encouraged to include BRD4 and MED1 ChIP data and H3K27ac to annotate the super-enhancers in myeloma.

Response 1

The work by Loven et al. (2013) reports the identification of multiple myeloma (MM) super-enhancers using MED1 in MM1.S and the dependence on BRD4 co-localisation for their activity, however to our knowledge there is no corresponding dataset in KMS-11.

We have re-processed and analysed the MM1.S BRD4 and MED1 ChIP-seq published by Loven et al. (2013) and overlapped these with the MM1.S SEs called from our own H3K27ac data. The same methodology was used for mapping, peak calling and SE calling (ROSE) as for the KMS-11 analyses presented in the manuscript.

We found MM1.S SEs are significantly enriched in p52, BRD4 and MED1 peaks and combinations thereof ($p < 2.2e-16$; Fisher’s exact test, odds ratios presented in Table 1). 83% of p52 bound SEs were bound by BRD4/MED1 including the SEs (RGS1-SE and BCL2-SE) characterised in the manuscript (see Figure 1).

Table 1. Transcription factor enrichment in super-enhancers

SE Enrichment	p52	BRD4	MED1	p52/BRD4/MED1
p52	1.67	1.85	1.71	2.91
BRD4		2.03	1.77	
MED1			1.80	

Figure 1. p52, BRD4 and MED1 binding across enhancers and super-enhancers (a-b) Euler diagrams displaying typical (TE) or super (SE) enhancer proportions bound by p52, BRD4 or MED1 factors. (c-d) Genome browser visualisation for the *BCL2* and *RGS1* loci encompassing proximal SEs. Tracks display: p52 ChIP-seq signal and peak calls in green; H3K4me1 ChIP-seq signal in blue; ChIP-seq signal from control and p52 KD experiments in KMS-11 in brown and orange respectively; SEs (black rectangles) and dynamic H3K27ac peaks (orange = lost) called from KMS-11 experiments; loops are indicated by arc annotation on the control and p52 KD ChIP-seq tracks and coloured by counts per million; BRD4 and MED1 ChIP-seq signals in MM1.S in dark red; ChIP-seq signal from control and p52 KD experiments in MM1.144 in brown and orange respectively; SEs (black rectangles) and constituent H3K27ac peaks classified by STABLO; gene track.

As mentioned in the manuscript discussion, our findings call for further exploration of p52-interacting co-factors in future studies. While this preliminary analysis relating to BRD4/MED1 in particular is interesting we think it deserves to be developed and featured in a separate follow up study rather than inflate this substantial communication. Nevertheless, we have added a reference to Loven et al. (2013) to the discussion (line 448) as this is a highly relevant piece of prior work that was inadvertently missed during preparation of our manuscript.

Minor comments

Comment 2

1. Page 3, line 100. Please change 'genomic expression' to mutational profile.

Response 2

Thank you, we have corrected this sentence (line 100-101).

Comment 3

2. Page 5, lines 199-203: It is unclear what the authors meant by "enhancer activity during myelomagenesis". To my understanding, MM develops from the MGUS stage (Monoclonal

gammopathy of undetermined significance) and transitions through SMM (smoldering multiple myeloma) to attain the MM stage. Since the authors are not capturing the developmental stages of the disease, I am unsure how justified or correct introducing a new classification system, STABILO, is.

Response 3

The classification proposed is not designed to recapitulate multiple myeloma development but rather highlight which enhancers may be specific to multiple myeloma or “borrowed” from B-cell stages where NF- κ B plays critical roles in directing B-cell development. We have edited the sentence to reflect this more clearly by omitting the mention of myelomagenesis (line 202).

Comment 4

3. Page 5, line 211: Are de novo and reactivated enhancers the same? What do the reactivated enhancers mean? Similarly, what does “co-opted from enhancers in B cells” mean? MM is a neoplasm of plasma B cells, not primary B cells. The heterogeneity is far more complicated than the way it is mentioned in the manuscript.

Response 4

De novo and reactivated enhancers are separate classes under the STABILO classification. As explained in lines 203-210 and Supp. Figure 2g, de novo enhancers are enhancers that show activation unique to multiple myeloma and never activated in B-cells (plasma or other). On the other hand, reactivated enhancers are enhancers that showed activation in multiple myeloma and non-plasma B-cells but not in plasma B-cells. Enhancers activated both in plasma B cells and multiple myeloma are named preserved enhancers.

Comment 5

4. Page 6, line 221: Here, authors mention that “p52 loci were frequently associated with de novo and reactivated enhancers”. This is highly confusing and confounding to the critique #3.

Response 5

We hope the changes to the previous comments have helped clarify this, however to be clear this line states that p52 binding sites were often detected within enhancers labelled as either “de novo” or “reactivated”. We have switched “and” to “or” to help highlight these are separate classes (line 224).

Comment 6

5. In figures, the panels that represent CHIP data, it would be nice to clarify the control in the legends.

Response 6

As far as we can see, the panels presenting CHIP-seq data are already annotated with “control” or abbreviated as “ctrl”.

Reviewer #2 (Remarks to the Author):

The authors have addressed my comments in detail, and revised the manuscript accordingly. I do not have any further comments and recommend accepting the manuscript.

Reviewer #3 (Remarks to the Author):

The authors successfully addressed to comments/suggestions and the manuscript is now acceptable.

Final remarks

We thank all reviewers for their thorough and helpful comments improving this manuscript.

Reviewers' Comments:

Reviewer #1:

None